# VCG-Bench: Towards A Unified Visual-Centric Benchmark for Structured Generation and Editing

**Xiaoyan Su** [* 1] **Peijie Dong** [* 1] **Zhenheng Tang** [2] **Song Tang** [1] **Yuyao Zhai** [3] **Kaitao Lin** [4] **Liang Chen** [3] **Yuhang Gai** [3] **Yuyu Luo** [1] **Qiang Wang** [5] **Xiaowen Chu** [1]

## Abstract

Despite the rapid advancements in Vision-Language Models (VLMs), a critical gap remains in their ability to handle structured, controllable diagrammatic tasks essential for professional workflows. Existing methods predominantly rely on pixel-based synthesis, which operates in probabilistic pixel spaces and is inherently limited in editability and fidelity. Instead, we propose a new "Diagram-as-Code" paradigm with symbolic logic that leverages `mxGraph` Extensible Markup Language (XML) for precise diagram generation and editing. We present **VCG-Bench**, a unified benchmark for visual-centric `mxGraph` tasks. VCG-Bench comprises: (1) a taxonomized dataset of 1,449 diverse diagrams spanning 6 domains and 15 sub-domains, (2) a paradigm definition that integrates Generation (Vision-to-Code) and Editability (Code-to-Code), (3) a Tailored Evaluation Protocol employing multi-dimensional metrics such as `mxGraph` Execution Success Rate, Style Consistency Score (SCS), etc. Experimental results highlight the challenges faced by current State-of-the-Art (SOTA) VLMs in structured fidelity and instruction compliance, reflecting their vision and reasoning capabilities.

## 1. Introduction

Vision-Language Models (VLMs) (OpenAI, 2023; Google DeepMind, 2025; Wu et al., 2025a; Chen et al., 2025b; Liu et al., 2023; Wang et al., 2024a; Bai et al., 2025) have demon-

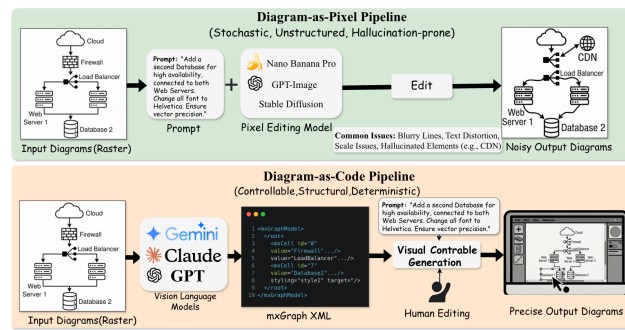

*Figure 1.* **Comparison of diagrammatic tasks.** VCG utilizes symbolic `mxGraph` XML for precise generation and editing, overcoming structural drift in pixel-based models.

*Table 1.* **Comparative analysis of visual representation formats.** `mxGraph` balances semantic richness with structured editability.

| Dimension | mxGraph | SVG | HTML/CSS | PPT |
|---|---|---|---|---|
| Primitive | Semantic Entities (Nodes, Edges) | Vector Paths (Lines, Curves) | DOM Elements (Tags, Styles) | Prop. Objects (Slides, Shapes) |
| Semantic Depth | **High** (Logical Topology) | Low (Visual Geometry) | Medium (Structure) | Low (Presentation) |
| Editability | **Structured** (Topology Preserved) | Unstructured (Pixel/Vector) | Rigid (Flow) | Restricted (GUI-centric) |
| Interpretability | **Open XML** | Open XML | Open Text | Prop./Binary |
| Controllability | **High** | Low | Medium | Poor |
| Modification Cost | **Low** (Software Ops) | High (Path Recalc.) | Medium (DOM/Style) | Low (GUI Manual) |

strated impressive capabilities in general multimodal tasks, spanning visual question answering (Antol et al., 2015) to open-ended image generation (Rombach et al., 2022; Podell et al., 2024). Through high-capacity architectures and massive-scale multimodal pre-training, modern VLMs have achieved near-human performance in scene description, optical character recognition (OCR), and cross-modal reasoning. Yet, moving beyond natural images to abstract, symbolic, and structured visual information remains a key frontier. This gap is particularly evident for diagrams such as flowcharts, system architectures, scientific figures, and UI mockups—which are pervasive in professional settings (Pan et al., 2024). Unlike natural images, diagrams are designed to encode precise semantics through composition, layout, and discrete symbols, requiring models to capture structure rather than appearance.

We propose **VCG-Bench**, a unified benchmark designed to

---

[*]Equal contribution [1]The Hong Kong University of Science and Technology (GuangZhou) [2]The Hong Kong University of Science and Technology [3]Huawei Technologies Co., Ltd [4]South China University of Technology [5]Harbin Institute of Technology, Shenzhen. Correspondence to: Zhenheng Tang <zhtang.ml@ust.hk>, Qiang Wang <qiang.wang@hit.edu.cn>, Xiaowen Chu <xwchu@hkust-gz.edu.cn>.

*Proceedings of the 43rd International Conference on Machine Learning*, Seoul, South Korea. PMLR 306, 2026. Copyright 2026 by the author(s).

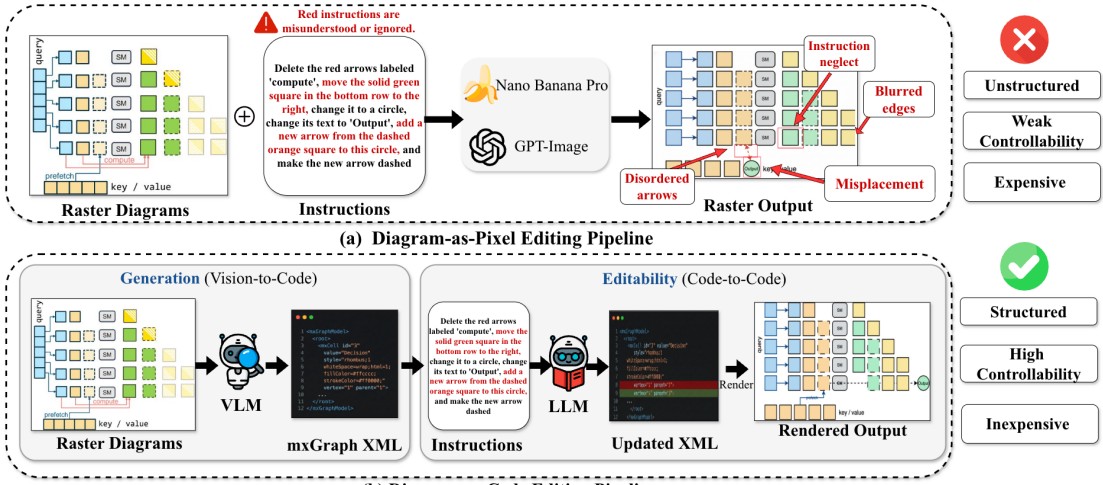

*(a) Diagram-as-Pixel Editing Pipeline*

*(b) Diagram-as-Code Editing Pipeline*

*Figure 2.* **Overview of the VCG-Bench framework.** Unlike fragmented approaches (top), VCG-Bench (bottom) unifies Vision-to-Code Generation (Task 1) and Instruction-to-Patch Editing (Task 2). Utilizing symbolic `mxGraph` XML enables precise, low-cost modifications for professional workflows.

evaluate the end-to-end capability of VLMs in generating and editing `mxGraph` XML diagrams. VCG-Bench follows a "Data-Task-Evaluation" framework. (1) We construct a diverse dataset categorized into 6 major domains (Academic, Software, Business, etc.), covering complex, logic-intensive layouts. (2) We unify diagrammatic tasks into two complementary streams: *Generation* (creating valid `mxGraph` XML from standard images) and *Editability* (modifying existing structures based on instructions), with an *Incremental Modification Strategy* to enhance efficiency. (3) We establish a rigorous evaluation protocol beyond simple text matching, incorporating execution success rates, visual style consistency (SCS), and semantic fidelity checks via QA. Our contributions are summarized below:

- We introduce VCG-Bench, the first benchmark bridging the gap between vision and editable structured generation.

- We curate a high-quality, taxonomized dataset of 1,449 diagrams with provenance and structural diversity.

- We propose a comprehensive evaluation protocol with novel metrics for executability, style consistency, and instruction following, benchmarking SOTA VLMs on diagrammatic tasks.

To motivate this benchmark design, we first contrast two diagrammatic paradigms. For diagrammatic tasks, one of the main paradigms is **"Diagram-as-Pixels"**. It relies on rasterized pixel generation and editing pipelines (e.g., prompt + pixel-editing models such as GPT-Image and diffusion models). Although it can produce visually plausible results, it is inherently **stochastic, unstructured, and hallucination-prone**, because it operates on pixels rather than symbolic structure. Fig. 1 illustrates that this often leads to *blurry lines, text distortion, scale/alignment errors,* and even *hallucinated elements* (e.g., inserting an unintended "CDN").

These issues conflict with practical diagram editing workflows that require: (1) **vector-accurate outputs** (e.g., SVG or `mxGraph`), (2) **low-cost incremental modifications**, and (3) **frequent iterative edits** without degradation.

To address these limitations, we adopt a **"Diagram-as-Code"** paradigm, which is more **controllable, structured, and deterministic**. Instead of editing pixels, we use **VLMs** (e.g., Gemini (Google DeepMind, 2025), Claude (Anthropic, 2025), and GPT (OpenAI, 2025)) to convert raster figures into structured representations. Guided by user instructions, the system performs edits via **Visual Controllable Generation (VCG)** with **symbolic operations**, enabling effective collaboration between LLMs and human users while preserving precise, editable diagram structure—producing **clean, high-fidelity** output figures (Fig. 1). Notably, the **Diagram-as-Code** paradigm can be integrated with pixel-level generators to produce more aesthetically pleasing, stylized renderings.

For **"Diagram-as-Code"**, Tab. 1 compares dominant representation formats (`mxGraph`, SVG, HTML/CSS, PPT). SVG is largely geometry-centric with limited semantics, HTML/CSS offers only moderate structural control, and PPT is GUI-centric with restricted programmatic editability. By contrast, `mxGraph` models diagrams as semantic entities (nodes/edges), providing **high semantic depth**, **structured editability**, and **high AI controllability** through an interpretable open XML representation, making it well-suited for AI-driven reasoning and precise modification. (See App. D.2 for further discussion.)

Fig. 3 details the data generation pipeline for VCG-Bench and shows the systematic process from data collection to structured evaluation. App. B provides detailed descriptions of the data synthesis pipeline, including Stage 1 (Image-to-

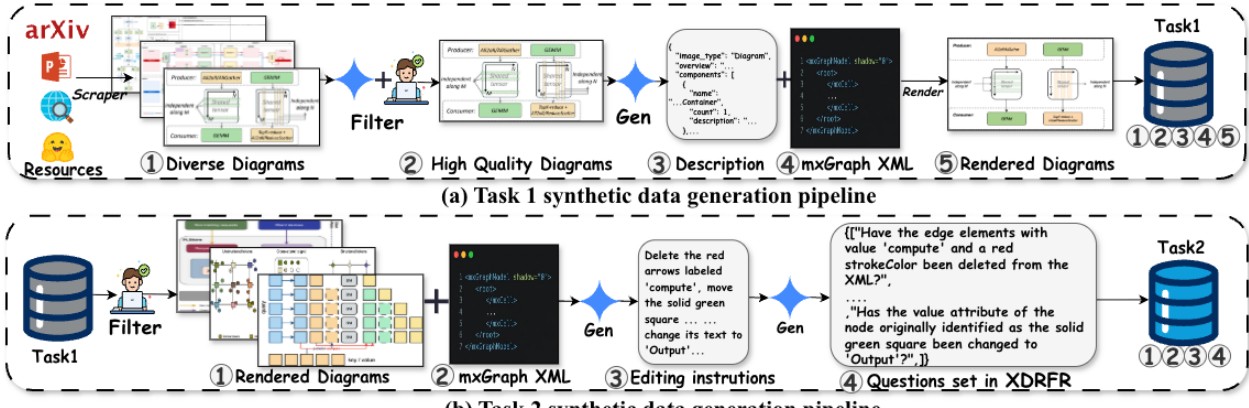

**(a) Task 1 synthetic data generation pipeline**

**(b) Task 2 synthetic data generation pipeline**

*Figure 3.* Overview of the VCG-Bench data generation framework. Subfigure (a) illustrates the end-to-end pipeline for Task 1, from raw web scraping to structured XML-based rendering. Subfigure (b) details the Task 2 pipeline, which focuses on generating editing-based reasoning tasks derived from Task 1.

mxGraph XML) and Stage 2 (Instruction Synthesis).

## 2. Related Work

Our work advances the domain of multimodal code generation by focusing on the mxGraph format. We situate it in four strands of related literature: chart and scientific plot generation, UI and symbolic graphics, instruction-driven image editing, and evaluation methodologies.

**Chart and Scientific Plot Generation.** Diagrammatic reasoning has been advanced by VQA benchmarks (e.g., AI2D (Kembhavi et al., 2016), ChartQA (Masry et al., 2022), FigureQA (Kahou et al., 2018), MMBench (Liu et al., 2024b)) and chart-understanding evaluations (CharXiv (Wang et al., 2024b), MMSCI (Li et al., 2024c), AIBench (Liao et al., 2026)). Recent work on executable code includes ChartMimic (Yang et al., 2025; Niu et al., 2025), Plot2Code (Wu et al., 2025b), PlotCraft (Zhang et al., 2025), MMCode (Li et al., 2024b), Flow2Code (He et al., 2025), and RealChart2Code (Zhang et al., 2026); most of this line remains centered on rasterized, data-driven visualizations. Interactive visualization systems such as HAIChart (Xie et al., 2024) study human-AI paired chart creation. Related to scientific diagrams, Draw with Thought (Cui et al., 2025) reconstructs static scientific diagrams into editable mxGraph XML. VCG-Bench provides a multi-domain Draw.io/mxGraph benchmark that jointly covers image-to-XML generation and instruction-based XML editing with explicit spatial and topological evaluation. Tab. 2 positions VCG-Bench against these benchmarks.

VCG-Bench is the only **multi-domain** benchmark that jointly supports **Edit.** and **Fine-grained** evaluation (Tab. 2). Prior visual-code benchmarks focus on single-domain generation (execute-pass or similarity); PlotCraft (Zhang et al., 2025) supports instruction-based editing and fine-grained

evaluation but remains within data visualization. We unify multi-domain diagram coverage with Task 1 (image-to-mxGraph XML) and Task 2 (instruction-based editing), evaluated via XQA, XDRFR, ESR, and SCS; this design supports both rigorous evaluation and scalable generation of verified visual-code pairs for model training.

**User Interface and Symbolic Graphics Generation.** UI-to-code has evolved from early CNN/RNN systems (Pix2Code (Beltramelli, 2018), Sketch2Code (Jain et al., 2019)) to diffusion-based generation (Rombach et al., 2022; Zhang et al., 2023b) and benchmarks such as Design2Code (Si et al., 2025); code LLMs (Chen et al., 2021; Li et al., 2023; Lozhkov et al., 2024; Roziere et al., 2023; Guo et al., 2024) have advanced program synthesis. In vector graphics, VCode (Lin et al., 2025) and SVGenius (Chen et al., 2025a) target SVG; learned (DeepSVG (Carlier et al., 2020), Im2Vec (Reddy et al., 2021)) and LLM-based (Jain et al., 2023; Vinker et al., 2022; Wu et al., 2023) methods address SVG. Recent benchmarks further evaluate SVG editing (Nishina & Matsui, 2024; 2025) and text-to-diagram generation/editing (Wei et al., 2025). These adjacent settings focus on SVG/vector graphics or general structured visuals; we target editable mxGraph XML with spatial and syntactic constraints, which preserves higher-level diagram topology and edit semantics for precise component manipulation.

**Instruction-Driven Image Editing.** Diffusion-based editing (Prompt-to-Prompt (Hertz et al., 2023), InstructPix2Pix (Brooks et al., 2023), MagicBrush (Zhang et al., 2023a)) has achieved significant progress but suffers from semantic entanglement (difficulty preserving unedited regions) and insufficient precision (character distortion, topological hallucination). We reframe editing as code refactoring on mxGraph XML, preserving exact fidelity outside edited regions and enabling edits that are infeasible in pixel space. This code-mediated

*Table 2.* Comparison of VCG-Bench with related benchmarks. **Edit.**: support for instruction-based editing. **Fine-grained**: multi-dimensional evaluation beyond single pass/fail or similarity. ✓: presence; ×: absence.

| Benchmark | Category | # of Test Instances | Edit. | Fine-grained | Metric |
|---|---|---|---|---|---|
| MMCode (Li et al., 2024b) | Visualization | 3,548 | × | × | Execute Pass |
| ChartMimic (Yang et al., 2025) | Chart | 4,800 | × | ✓ | Similarity |
| Plot2Code (Wu et al., 2025b) | Scientific Plot | 368 | × | × | Execute Pass |
| Design2Code (Si et al., 2025) | Web UI | 484 | × | ✓ | Similarity |
| VCode (Lin et al., 2025) | SVG | 464 | × | × | RenderVQA |
| PlotCraft (Zhang et al., 2025) | Data Vis. | 982 | ✓ | ✓ | Multi-Level |
| CharXiv (Wang et al., 2024b) | Chart | 2,323 | × | × | Accuracy |
| MM-Vet (Yu et al., 2023) | General | 218 | × | × | OpenQA |
| MMBench (Liu et al., 2024b) | General | 3,217 | × | × | MCQ |
| **VCG-Bench (Ours)** | **Multi-domain** | **1,449** | ✓ | ✓ | **XQA/XDRFR/ESR/SCS** |

paradigm thus avoids the semantic entanglement and precision limits of pixel-level editing while supporting deterministic, localized modifications.

**Evaluation Methodologies for Instruction Following.** Robust evaluation remains a critical challenge; traditional metrics often fail to capture the nuance of complex instructions. InFoBench (Qin et al., 2024; Zhou et al., 2023) introduced the Decomposed Requirements Following Ratio (DRFR); multimodal benchmarks (Yu et al., 2023; Yue et al., 2024; 2025; Li et al., 2024a; Xu et al., 2024; Chen et al., 2024) and domain-specific work (Yang et al., 2025; Wu et al., 2025b; Si et al., 2025; Zhuo et al., 2025; Xie et al., 2026) use fine-grained criteria, including structured-visual factuality checks and visualization-quality assessment. Drawing inspiration from these, we adopt a decomposed strategy (XDRFR and related metrics) for structural and visual fidelity of Draw.io outputs, allowing fine-grained verification of instruction compliance directly on the generated XML. Evaluating on the XML structure rather than rendered images avoids visual ambiguities and supports reproducible, attribute-level checks. A manual audit of XDRFR (App. A) confirms high agreement between automated scores and human judgment.

## 3. VCG-Bench Construction

### 3.1. Data Provenance & Licensing

Our dataset is curated from four sources: Open-source Template Repositories (40%), Open Access Academic Papers (30%), Anonymized Corporate Diagrams (20%), and Permissively Licensed Web Diagrams (10%). We incorporate the **SALT-NLP/Design2Code** dataset (Si et al., 2025) from Hugging Face Hub under its original licensing terms. The dataset, comprising mxGraph XML source files, rendered high-resolution images, captions, and editing instructions, is released under **CC BY 4.0**. The evaluation toolkit is released under **MIT License**. Web-crawled data undergoes PII redaction to remove names, IP addresses, and sensitive corporate identifiers. App. B provides details on data sources, licensing policies, and diversity considerations.

*Table 3.* **Dataset Taxonomy.** Summary of the 6 major domains and 15 sub-domains.

| Domain (L1) | Sub-domains (L2) | Key Characteristics |
|---|---|---|
| **Academic** | Arch., Neural Nets, Data Viz. | Scientific research & logic |
| **Software** | UML, Sequence, ER Diagram | Standardized modeling |
| **Business** | Product, Report, Strategy | Planning & operations |
| **Management** | Gantt, Hierarchy, Flow | Collaboration & SOPs |
| **UIUX** | Web, Mobile | Interaction design |
| **General** | Mind Map | Logical thinking |

### 3.2. Quality Assurance Pipeline

We employ a "Machine-Generation, Human-Verification" pipeline: (1) **Acquisition & Captioning**: Gemini-3-Pro generates structured JSON captions of visual elements and layout; (2) **mxGraph XML Synthesis**: during data construction, candidate mxGraph XML files are synthesized from image-caption pairs by multiple VLM candidates and retained only after downstream quality filters; (3) **Parser Validation**: rule-based validation against the mxGraph schema filters errors; (4) **Rendering Verification**: similarity between rendered $I_{gen}$ and source $I_{src}$ is computed via SigLIP2 (Tschannen et al., 2025) embeddings, discarding samples with cosine similarity $< 0.85$; (5) **Human Review**: annotators verify structural completeness, text accuracy ($> 95\%$ OCR match), and editability. This data-construction process is separate from the formal benchmark evaluation in Sec. 6, where all evaluated models are run on the released Task 1/Task 2 inputs. From 5,000+ candidates, parsing and rendering success rates were 60% and 55%, respectively. After human filtering (45% acceptance), 1,449 high-quality samples were retained, requiring 2.3 edits per sample on average for alignment.

### 3.3. Dataset Taxonomy

The benchmark categorizes 1,449 examples into 6 major domains (L1) and 15 sub-domains (L2), ensuring broad coverage of diagrammatic complexity. App. B.1 provides a detailed breakdown, with summary in Tab. 3; Fig. 4 visualizes distribution across sub-domains and difficulty.

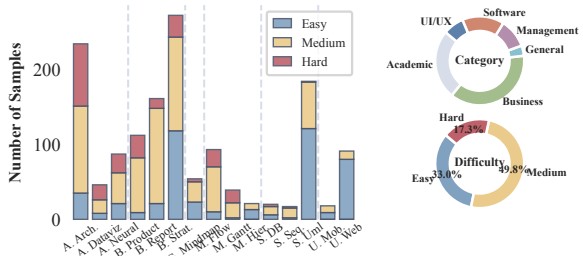

*Figure 4.* Left: **Distribution across 15 sub-domains.** Stratified by difficulty, reflecting structural complexity and element density. Right: **Dataset composition.**

## 4. Task Definition

We define two core tasks under a unified framework formalized by **executability constraints**: both require model outputs to be valid `mxGraph` XML that can be parsed and rendered. The following subsections formalize Generation (Vision-to-Code) and Editability (Code-to-Code).

### 4.1. Generation (Vision-to-Code)

**Definition**: Given a raster image $I \in \mathbb{R}^{H \times W \times 3}$ and a structured caption $S$ of visual elements and layout, the model generates a valid **mxGraph XML** string $C$ such that $Render(C)$ approximates $I$ in structure and semantics. **Optimization Goal**: The model maximizes likelihood $p_\theta(C|I, S)$. **Constraints**: ① **Syntactic Validity**: $C$ must be valid XML parsable by the `mxGraph` library. ② **Visual Fidelity**: $Render(C)$ must match $I$ in topological structure (node connections), spatial layout (bounding boxes), and textual content. **Output**: A raw `mxGraph` XML string containing 'mxGraphModel' definitions. App. D provides the `mxGraph` XML schema and format details. Our main Task-1 setting uses image plus structured caption as input; Sec. 6.2 reports an image-only ablation showing that raw-image-to-`mxGraph` is substantially harder.

### 4.2. Editability (Code-to-Code)

**Definition**: Given source `mxGraph` XML $C_{src}$, rendering $I_{src}$, and instruction $T$ (e.g., "Change the decision node from red to blue"), the model predicts a **Differential Patch** $P$, yielding $C_{tgt} = Apply(C_{src}, P)$. **Incremental Modification Strategy**: Rather than regenerating the full `mxGraph` XML, which is token-expensive and prone to structural drift, we predict only the `mxGraph` XML fragment pairs to be changed. **Patch Schema**: We use a deterministic `JSON`-based replacement format, where each change contains original and modified `mxGraph` XML fragments:

```
1  {
2    "changes": [{
3      "original_fragment": "<mxCell id=\"s2_r0\" ...
4        fillColor=#FF0000 ...>...</mxCell>",
5      "modified_fragment": "<mxCell id=\"s2_r0\" ...
6        fillColor=#0000FF ...>...</mxCell>"
7    }]
8  }
```

**Patch Applier**: A deterministic applier $\Phi(C, P)$ replaces the matched source fragment in $C$ with the target fragment; deletion uses an empty target. If exact matching fails, fuzzy regex matching ignores whitespace.

### 4.3. Instruction Taxonomy

We develop an instruction generation taxonomy grounded in **atomic operations**. We define a set of **14 atomic operation categories** that encompass diagram editing tasks, organized into four domains: (1) **Node Attribute Modification** (4 types): altering color, shape, size, and text; (2) **Node Structure Operations** (3 types): insertion, deletion, and relocation; (3) **Edge Attribute Modification** (3 types): adjusting color, style, and arrow endpoints; and (4) **Edge Structure Operations** (4 types): insertion, deletion, redirection, and path updates.

**Complexity Stratification.** We stratify editing instructions into three tiers by the number of atomic operations required: **Easy** (1–2 operations), **Medium** (3–4 operations), and **Hard** (5–7 operations). This hierarchical design ensures a robust evaluation across varying modification intensity. For each sample, we generate instructions for all three tiers. Instruction synthesis is automated via LLMs, conditioning on the rendered images and the `mxGraph` XML structure. To mirror real-world usage, instructions employ natural language descriptors (e.g., referencing node text) rather than technical identifiers. App. E provides prompt templates for instruction generation, XML editing, and evaluation.

## 5. Evaluation Protocol

We establish an evaluation protocol measuring three dimensions: (1) **Executability**, assessing syntactic validity and renderability; (2) **Visual Fidelity**, verifying structural and semantic reconstruction; and (3) **Semantic Compliance**, evaluating instruction adherence and logic preservation. All model-based evaluations employ **Gemini-3-Pro** for consistency. Among our metrics, SCS is the most exposed to judge-family effects because it relies on visual judgment, whereas ESR is programmatic and CodeXQA/XDRFR operate over structured XML-based QA or verification. We therefore report a cross-judge validation for SCS in Tab. 5. Metrics are summarized in Tab. 4. App. F provides detailed mathematical formulations, question set construction procedures, and validation protocols.

*Table 4.* **Metrics Overview.** Lists all evaluation metrics used in VCG-Bench along with their definitions and inputs.

| Metric | Task | Definition | Input |
|---|---|---|---|
| **ESR** | 1/2 | Execution Success Rate | Generated `mxGraph` XML |
| **SCS** | 1/2 | Style Consistency Score | Original and rendered images |
| **CodeXQA** | 1 | Semantic Fidelity | Generated `mxGraph` XML and questions |
| **SigLIP2** | 1 | Visual similarity | Original and rendered images |
| **XDRFR** | 2 | XML Decomposed Require-ments Following Ratio | Model output `XML` fragments |

*Figure 5.* Performance scaling and robustness across diagrammatic complexity tiers. Each panel illustrates the CodeXQA accuracy of a specific model family as task difficulty increases from Easy to Hard.

*Table 5.* **Cross-judge SCS validation.** Entries report mean $\pm$ standard deviation on the shared valid Task-1 subset ($n = 78$).

| Generator Model | Gemini Judge | GPT Judge | Claude Judge |
|---|---|---|---|
| GPT-5.2 | $0.626 \pm 0.208$ | $0.688 \pm 0.121$ | $0.635 \pm 0.172$ |
| Gemini-3-Pro | $0.788 \pm 0.148$ | $0.785 \pm 0.095$ | $0.754 \pm 0.105$ |
| Claude-4.5-Sonnet | $0.705 \pm 0.186$ | $0.748 \pm 0.118$ | $0.690 \pm 0.144$ |
| GLM-4.6V | $0.415 \pm 0.184$ | $0.572 \pm 0.153$ | $0.465 \pm 0.182$ |

**Executability Metric.** **Execution Success Rate (ESR).** We check whether generated code is syntactically valid and renderable. For code $C$, ESR indicates successful parsing by the `mxGraph` engine and rendering into image $I_{gen}$:

$$\text{ESR} = \mathbb{I}(\text{Parsable}(C) \wedge \text{Renderable}(C)) \qquad (1)$$

where $\mathbb{I}(\cdot)$ is the indicator function. Samples with $\text{ESR} = 0$ receive zero on subsequent metrics, serving as a filter.

**Visual Fidelity Metrics.** **Style Consistency Score (SCS).** SCS uses a VLM-based evaluator (Gemini-3-Pro) to assess visual alignment, addressing limitations of pixel-level metrics in capturing structural aesthetics. For **Task 1 (Generation)**, SCS evaluates three dimensions on a 10-point scale: (1) *Visual Style* (color, stroke, shape); (2) *Layout Consistency* (spatial topology); (3) *Aesthetic Quality* (alignment, balance). Each score is prompted, averaged, and normalized to $[0, 1]$. For **Task 2 (Editing)**, SCS focuses on *Style Consistency* and *Aesthetic Quality* relative to pre-edit, ensuring modifications preserve professional standards.

**Robustness of SCS to judge choice.** Since SCS is a judge-based visual metric, we additionally test whether its relative conclusions are stable across evaluator families. We randomly sample 100 Task-1 examples, fix the outputs from four representative generators (GPT-5.2, Gemini-3-Pro, Claude-4.5-Sonnet, and GLM-4.6V), and re-score the same outputs with Gemini, GPT, and Claude judges. We retain the shared valid subset where all judges return directly comparable scores ($n = 78$).

Tab. 5 shows noticeable judge calibration differences in absolute SCS values, especially for weaker outputs. However, all three judges produce the same model ranking: Gemini-3-Pro > Claude-4.5-Sonnet > GPT-5.2 > GLM-4.6V. Pairwise Spearman correlations are also consistent (0.770 for Gemini–GPT, 0.735 for Gemini–Claude, and 0.721 for GPT–

Claude; average 0.742). Thus, we interpret SCS primarily as a relative visual-quality signal: its absolute values are judge-calibrated, but the ranking conclusions used in our analysis are stable across evaluator families.

**SigLIP2 Visual Similarity.** To capture high-level structural and layout alignment, we compute cosine similarity between embeddings of rendered image $I_{gen}$ and ground truth $I_{gt}$ using SigLIP2 (Tschannen et al., 2025):

$$S_{\text{vis}} = \cos(E_{\text{img}}(I_{gen}), E_{\text{img}}(I_{gt})) \qquad (2)$$

where $E_{\text{img}}$ denotes the image encoder. This approach captures high-level structural and layout semantics more robustly than pixel-level metrics.

**Semantic Compliance Metrics.** **CodeXQA (Task 1).** We introduce **CodeXQA** to evaluate whether generated `mxGraph` XML topologically preserves source image information. We generate question sets $Q = \{q_{\text{cnt}}, q_{\text{id}}, q_{\text{rel}}\}$ covering three cognitive levels: (1) **Counting**: enumerating elements; (2) **Identification**: retrieving attributes; and (3) **Relationship**: reasoning about connectivity. The metric measures accuracy in answering questions based solely on the generated `mxGraph` XML code.

**XDRFR (Task 2).** We propose **XDRFR (XML-based DRFR)**, adapted from DRFR (Qin et al., 2024), to evaluate instruction following in code editing. Each editing instruction $T$ is decomposed into atomic verification questions $Q_T = \{q_1, \ldots, q_n\}$. Evaluation operates directly on the modified `mxGraph` XML structure rather than rendered images, leveraging structured attribute information for precise logic verification and avoiding visual ambiguities. The score is the ratio of satisfied requirements:

$$\text{XDRFR} = \frac{1}{|Q_T|} \sum_{i=1}^{|Q_T|} \mathbb{I}(\text{Verify}(C_{tgt}, q_i)) \qquad (3)$$

We conducted a manual spot-check of 55 samples for the XDRFR metric (App. A, Tab. 11), confirming high reliability with only one minor issue requiring manual correction in a single sample.

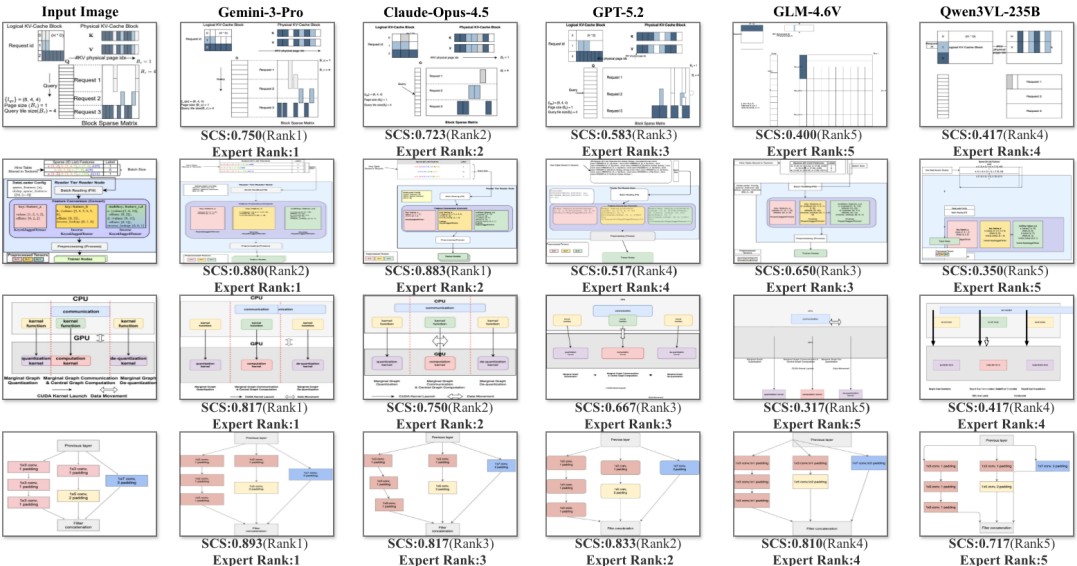

*Figure 6.* **Task 1 qualitative examples.** This figure showcases generated diagrams across multiple domains with their corresponding Style Consistency Scores (SCS), SCS rankings, and expert rankings. The examples demonstrate that the SCS metric rankings and human expert rankings are highly consistent, proving that the SCS metric accurately reflects human aesthetic standards.

## 6. Experiments

### 6.1. Experimental Setup

We evaluate leading multimodal models on VCG-Bench for baselines. For all models we use temperature 0 and a maximum token limit (e.g., 4096) for deterministic outputs. For Task 1, Gemini-3-Pro generates structured descriptions per image to assist `mxGraph` XML generation; for Task 2 (Editability), we provide `diff` patch schema in the prompt. We evaluate closed-source models (Gemini-3-Pro (Google DeepMind, 2025), Gemini-3-Flash (Google DeepMind, 2025), Claude-4.5-Opus (Anthropic, 2025), Claude-4.5-Sonnet (Anthropic, 2025), GPT-5.2 (OpenAI, 2025)) and open-source models (Qwen3-VL series 235B/32B/8B (Qwen Team, 2025a), Qwen3-Omni-30B (Qwen Team, 2025a), Qwen3-Coder (Qwen Team, 2025b), GLM-4.6V (GLM-V Team, 2025) (Task 1), GLM-4.6 (Task 2), DeepSeek-V3.2 (DeepSeek-AI, 2025), Kimi-K2 (Kimi Team, 2025), MiniMax-M2 (MiniMax-AI, 2025), Step3 (Huang & StepFun Team, 2026), among others). Additional results and breakdowns are in App. A.

### 6.2. Vision-to-XML Generation

Task 1 evaluates the capability of models to translate raster images into editable `mxGraph` XML code, serving as the gateway to the editing workflow and requiring strong perception and structural reasoning. We report overall results first, then break down by difficulty tier and by diagrammatic reasoning primitive. **Open-source models fail primarily on executability (ESR), though some retain moderate perceptual (CodeXQA) scores.** Tab. 7 gives overall re-

sults; Tab. 8 gives the breakdown by difficulty. App. A.1.1 further compares Task 1 with a representative pipeline baseline, showing that coarse layout recovery does not recover editable Draw.io structure.

**Input ablation.** We ablate the structured caption on 100 Task-1 examples to test whether it provides grounding beyond the image. Removing it lowers ESR from 1.000 to 0.980, SCS from 0.805 to 0.690, and CodeXQA from 0.930 to 0.786 (Tab. 6), confirming that captions help localize elements, layout, and relations for `mxGraph` conversion. We therefore use image-plus-caption as the main setting and image-only generation as a harder ablation.

*Table 6.* **Caption ablation for Task 1.** Results are measured on a random subset of 100 examples. The main setting uses image plus structured caption, while the ablation removes the caption and uses only the raw image.

| Setting | ESR | SCS | CodeXQA |
|---|---|---|---|
| With caption (main setting) | 1.000 | 0.805 | 0.930 |
| Without caption | 0.980 | 0.690 | 0.786 |

Tab. 7 shows a large gap between closed-source and open-source models. Open-source models often produce non-executable code (e.g., Qwen3-VL-32B/8B have ESR 0.0000), while closed-source models keep ESR above 0.84, indicating a lack of explicit Draw.io domain knowledge on the open side. High perceptual scores do not guarantee valid graph generation: `GLM-4.6V` reaches CodeXQA 0.9032 but only 0.7471 ESR and low `SigLIP2`, i.e., it detects elements but fails to synthesize valid structural code; this gap aligns with counting failures (Tab. 9). Gemini-3-Pro is state-of-the-art on style consistency (SCS **0.7790**); most open-source models fail on perception (Qwen3-VL-

*Table 7.* Overall benchmark results for Task 1: Vision-to-XML Generation. **Open models fail mainly on ESR, despite moderate CodeXQA for some.** Best scores in bold.

| C. | Model | SCS | CodeXQA | ESR | SigLIP2 |
|---|---|---|---|---|---|
| Closed | Claude-4.5-Opus | 0.6502 | 0.8947 | 0.9247 | 0.8769 |
| | GPT-5.2 | 0.5617 | 0.9255 | 0.8424 | 0.7974 |
| | Gemini-3-Pro | **0.7790** | **0.9313** | **0.9626** | **0.9162** |
| | Gemini-3-Flash | 0.7601 | 0.9263 | 0.9556 | 0.9101 |
| | Claude-4.5-Sonnet | 0.6447 | 0.9072 | 0.9606 | 0.8892 |
| Open | Qwen3-VL-235B | 0.1519 | 0.8224 | 0.3462 | 0.2808 |
| | Step3 | 0.0121 | 0.8293 | 0.0518 | 0.0374 |
| | Qwen3-VL-8B | 0.0000 | 0.1041 | 0.0000 | 0.0000 |
| | Qwen3-VL-32B | 0.0000 | 0.3514 | 0.0000 | 0.0000 |
| | GLM-4.6V | **0.3433** | **0.9032** | **0.7471** | **0.6562** |
| | Qwen3-Omni-30B | 0.1932 | 0.6472 | 0.6296 | 0.4792 |

32B/8B and Qwen3-VL-235B score 0.0000 and 0.1519), with `GLM-4.6V` leading open-source at 0.3433.

*Table 8.* Task 1 (Vision-to-XML Generation) performance breakdown across difficulty tiers.

| D. | C. | Model | SCS | CodeXQA | ESR | SigLIP2 |
|---|---|---|---|---|---|---|
| Easy | Closed | Claude-4.5-Opus | 0.7129 | 0.9430 | 0.9434 | 0.8983 |
| | | GPT-5.2 | 0.6169 | **0.9513** | 0.8532 | 0.8094 |
| | | Gemini-3-Pro | **0.8279** | 0.9370 | **0.9895** | **0.9438** |
| | | Gemini-3-Flash | 0.8094 | 0.9436 | 0.9789 | 0.9316 |
| | | Claude-4.5-Sonnet | 0.6931 | 0.9473 | 0.9623 | 0.9071 |
| | Open | Qwen3-VL-235B | 0.2268 | 0.9270 | 0.4528 | 0.3681 |
| | | Step3 | 0.0108 | 0.9170 | 0.0461 | 0.0334 |
| | | Qwen3-VL-8B | 0.0000 | 0.1641 | 0.0000 | 0.0000 |
| | | Qwen3-VL-32B | 0.0000 | 0.4704 | 0.0000 | 0.0000 |
| | | GLM-4.6V | **0.4342** | **0.9330** | **0.7925** | **0.7203** |
| | | Qwen3-Omni-30B | 0.2289 | 0.7374 | 0.6247 | 0.4846 |
| Medium | Closed | Claude-4.5-Opus | 0.6425 | 0.9169 | 0.9139 | 0.8705 |
| | | GPT-5.2 | 0.5551 | 0.9371 | 0.8458 | 0.8037 |
| | | Gemini-3-Pro | **0.7753** | **0.9401** | **0.9694** | **0.9218** |
| | | Gemini-3-Flash | 0.7608 | 0.9368 | 0.9583 | 0.9142 |
| | | Claude-4.5-Sonnet | 0.6508 | 0.9253 | 0.9653 | 0.8946 |
| | Open | Qwen3-VL-235B | 0.1289 | 0.8247 | 0.3042 | 0.2500 |
| | | Step3 | 0.0130 | 0.8520 | 0.0528 | 0.0387 |
| | | Qwen3-VL-8B | 0.0000 | 0.0936 | 0.0000 | 0.0000 |
| | | Qwen3-VL-32B | 0.0000 | 0.3309 | 0.0000 | 0.0000 |
| | | GLM-4.6V | **0.3236** | **0.9057** | **0.7444** | **0.6583** |
| | | Qwen3-Omni-30B | 0.1771 | 0.6684 | 0.6083 | 0.4659 |
| Hard | Closed | Claude-4.5-Opus | 0.5527 | 0.7131 | 0.9200 | 0.8545 |
| | | GPT-5.2 | 0.4752 | 0.8403 | 0.8120 | 0.7563 |
| | | Gemini-3-Pro | **0.6958** | **0.8948** | 0.8911 | 0.8469 |
| | | Gemini-3-Flash | 0.6642 | 0.8617 | 0.9040 | **0.8573** |
| | | Claude-4.5-Sonnet | 0.5347 | 0.7684 | **0.9440** | 0.8398 |
| | Open | Qwen3-VL-235B | 0.0755 | 0.5857 | 0.2640 | 0.2032 |
| | | Step3 | 0.0122 | 0.6491 | 0.0600 | 0.0411 |
| | | Qwen3-VL-8B | 0.0000 | 0.0435 | 0.0000 | 0.0000 |
| | | Qwen3-VL-32B | 0.0000 | 0.2030 | 0.0000 | 0.0000 |
| | | GLM-4.6V | **0.2269** | **0.8326** | 0.6680 | **0.5279** |
| | | Qwen3-Omni-30B | 0.1724 | 0.4135 | **0.7000** | 0.5076 |

Across Easy, Medium, and Hard levels (by `mxGraph` XML token count, Fig. 5), closed-source models like Gemini-3-Pro show controlled degradation (CodeXQA 0.9370→0.8948, $-4.5\%$), whereas Qwen3-VL-8B/32B collapse on Hard and Qwen3-VL-235B decays sharply ($-36.8\%$). Only `GLM-4.6V` stays relatively stable, though still below closed-source models.

### 6.2.1. ERROR ANALYSIS

We analyze Task 1 failures through the diagrammatic primitives in Tab. 9—**Counting**, **Identification**, and **Relationship**—which correspond to enumerating elements, recognizing types/attributes, and inferring connectivity in vision-

*Table 9.* Task 1 accuracy across diagrammatic reasoning primitives. Evaluates models' ability to parse structural elements from raster images.

| C. | Model | Accuracy by Question Type | | |
|---|---|---|---|---|
| | | Counting | Identification | Relationship |
| Closed | Claude-4.5-Opus | 0.8925 | 0.9035 | 0.8881 |
| | Claude-4.5-Sonnet | 0.9079 | 0.9209 | 0.8928 |
| | Gemini-3-Flash | 0.9361 | 0.9249 | **0.9102** |
| | Gemini-3-Pro | **0.9488** | 0.9299 | 0.9074 |
| | GPT-5.2 | 0.9212 | **0.9451** | 0.9100 |
| Open | Qwen3-Omni-30B | 0.6384 | 0.6680 | 0.6351 |
| | Qwen3-VL-235B | 0.8092 | 0.8599 | 0.7981 |
| | Qwen3-VL-32B | 0.3064 | 0.3981 | 0.3498 |
| | Qwen3-VL-8B | 0.1492 | 0.0794 | 0.0838 |
| | Step3 | 0.7989 | 0.8678 | 0.8212 |
| | GLM-4.6V | **0.8812** | **0.9326** | **0.8959** |

to-XML. This clarifies why open-source models lag on executability (Tab. 7) and difficulty tiers (Tab. 8).

*Counting as the dominant failure mode.* Counting is the weakness for open-source models. Qwen3-VL-8B reaches 0.1492 (vs. Gemini-3-Pro 0.9488); Qwen3-VL-32B 0.3064; Qwen3-VL-235B stays at 0.8092. This **instance-grounding** failure drives low SCS: without accurate counts, models cannot produce topologically correct graphs, explaining the gap between moderate CodeXQA and low ESR/SCS (e.g., `GLM-4.6V` in Tab. 7). Identification and Relationship show deficits but less severe for stronger—e.g., `GLM-4.6V` attains 0.9326 Identification and 0.8959 Relationship, near closed-source; Counting (0.8812) is the bottleneck.

*Open vs. closed and within-model patterns.* Closed-source models stay balanced across all three primitives ($\approx$0.89–0.95). Among open-source ones, Qwen3-VL-8B fails uniformly (0.08–0.15); Qwen3-VL-32B and Qwen3-VL-235B show Counting as the bottleneck. Step3 reaches 0.8678 Identification but only 0.7989 Counting, again highlighting instance grounding as the primary failure mode. Fig. 6 shows qualitative examples with SCS and expert rankings.

### 6.3. Instruction-based Diagram Editing

Task 2 evaluates the hypothesis that, once a diagram is available as structured code, constrained incremental editing becomes substantially more precise and controllable. We evaluate the same closed- and open-source models on instruction-based patch editing; all achieve high ESR and SCS, with XDRFR distinguishing instruction-following quality. Tab. 10 gives the overall results.

Under constrained XML patching, all models achieve high editing fidelity; Gemini-3-Pro leads with XDRFR **0.9405**. With ESR and SCS nearly saturated, **XDRFR** becomes the key discriminative signal (0.8873–0.9405). This indicates that frontier models are mature at localized edits given complete diagram structure, but not necessarily at

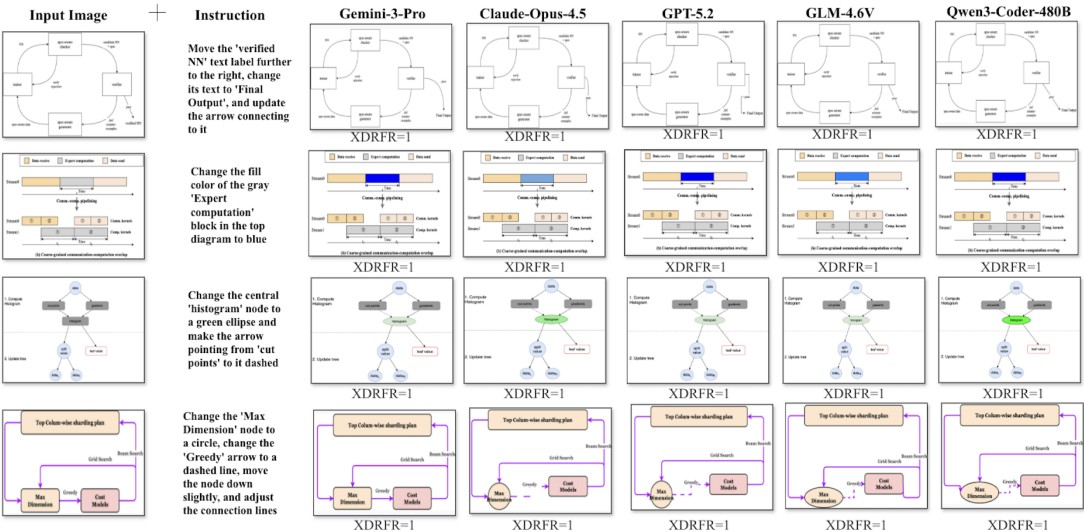

*Figure 7.* **Task 2 instruction editing precision demonstration.** Rows show distinct editing instructions; columns show the input diagram, instruction, and outputs from five representative models.

*Table 10.* Overall benchmark results for Task 2: Instruction-based Diagram Editing. **High performance across models suggests that constrained code-mediated editing is a robust paradigm.** Best scores in bold.

| C. | Model | ESR | SCS | XDRFR |
|---|---|---|---|---|
| Closed | Claude-4.5-Opus | 0.9986 | 0.9163 | 0.9271 |
| | GPT-5.2 | 0.9978 | **0.9163** | 0.9334 |
| | Gemini-3-Pro | **0.9993** | 0.9109 | **0.9405** |
| | Gemini-3-Flash | **0.9993** | 0.9151 | 0.9295 |
| | Claude-4.5-Sonnet | **0.9993** | 0.8954 | 0.9287 |
| Open | Kimi-K2 | 0.9964 | 0.9021 | 0.9276 |
| | GLM-4.6 | 0.9942 | 0.9011 | 0.9202 |
| | Qwen3-VL-235B | 0.9964 | 0.9055 | **0.9312** |
| | MiniMax-M2 | 0.9920 | 0.9004 | 0.9151 |
| | Qwen3-VL-32B | **0.9986** | **0.9295** | 0.8873 |
| | Qwen3-Coder | 0.9942 | 0.9076 | 0.9210 |
| | DeepSeek-V3.2 | 0.9957 | 0.9099 | 0.8928 |

open-ended redesign or ambiguous intent. Qwen3-VL-235B reaches 0.9312, ahead of some closed-source models, while Qwen3-VL-32B pairs high SCS (0.9295) with lower XDRFR (0.8873), suggesting stronger semantic preservation than detail precision. Fig. 7 gives qualitative examples; App. A.1.6 compares against a pixel-based editing baseline.

Fig. 7 covers representative edits: text/connector updates, color changes, shape/style transformation, and multi-operation edits over shape, position, line style, and connectors. Across model families, outputs preserve layout and style while applying local changes, explaining high SCS and why XDRFR remains the main discriminative metric. These cases also show XML patching's practical advantage: edits localize to explicit nodes, edges, and attributes rather than regenerated pixels, making results easier to verify and refine in Draw.io.

### 6.4. Discussion

Comparing Task 1 and Task 2 (Tabs. 7 and 10) reveals a sharp vision-to-code vs. code-to-code gap. Models such as **Qwen3-VL-32B** fail on Task 1 (generating `mxGraph` XML from images) yet achieve strong Task-2 editing fidelity. They can reason about XML structure and follow edits; the bottleneck is **visual understanding**—inferring diagram structure from pixels. VCG-Bench separates these capabilities and shows that models unable to produce code from images can still edit provided code effectively.

**Scale vs. performance.** The Qwen3-VL series (8B, 32B, 235B) shows diminishing returns: perceptual gains (CodeXQA: 0.1041→0.8224) outpace structured generation (ESR: 0.0000→0.3462, SCS: 0.0000→0.1519). `GLM-4.6V` outperforms larger models despite fewer parameters—**architecture and training data quality matter more than scale** for structured generation.

## 7. Conclusion

We have introduced **VCG-Bench**, the first unified benchmark designed for evaluating controllable structured generation in VLMs. By shifting the focus from passive perception to active, editable code generation, VCG-Bench exposes significant limitations in current state-of-the-art models in fine-grained visual parsing and syntactic constraint satisfaction. Our framework offers researchers a reproducible and quantifiable testbed to assess the practical readiness of multimodal agents for professional engineering workflows. We will release the benchmark and evaluation toolkit to support future work. Future work will expand the dataset to include dynamic interaction histories and multi-page diagrams.

## Acknowledgement

This paper was supported by the NSF of China (62402409); Youth S&T Talent Support Programme of Guangdong Provincial Association for Science and Technology (SKXRC2025461); the Young Talent Support Project of Guangzhou Association for Science and Technology (QT-2025-001); Guangzhou Basic and Applied Basic Research Foundation (2026A1515010269, 2025A04J3935, 2023A1515110545); and Guangzhou-HKUST(GZ) Joint Funding Program (2025A03J3714); Hong Kong CRF grants under Grant No. C7004-22G and C6015-23G.

## Impact Statement

This paper presents work whose goal is to advance the field of Machine Learning, specifically in the evaluation and development of Vision-Language Models (VLMs) for structured visual content generation. VCG-Bench introduces a standardized benchmark for assessing diagram generation and editing capabilities, which has broader implications for software engineering, technical documentation, and design automation workflows.

We believe the primary societal impact will be positive, as improved diagram generation capabilities can enhance productivity in technical fields, improve accessibility of visual information, and support educational applications. The open-source nature of our benchmark promotes transparency and reproducibility in research, while the standardized evaluation protocol helps identify and address model limitations before deployment.

## Limitations

VCG-Bench is limited to **Draw.io/mxGraph** and has not been validated on other diagram languages (e.g., Visio, TikZ, Mermaid) or in real-world deployment. The dataset may underrepresent non-English labels, rare styles, and specialized notations. The current editing task focuses on objectively verifiable, rule-based incremental edits; it does not fully capture semantic ambiguity, mixed-initiative clarification, or subjective diagram-level redesign. **Executability** does not guarantee correctness—diagrams can render yet contain semantic or topological errors, requiring expert review for high-stakes. Future work may extend to other formats, interactive clarification protocols, and real-world suites.

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

# Appendix Contents

## A. Experimental Results

### A.1. Additional Experimental Results

#### A.1.1. PAPER2ANY PIPELINE BASELINE FOR TASK 1

We additionally examine Paper2Any (OpenDCAI, 2025) as a representative non-LLM pipeline baseline for Task 1. This baseline is closer to an OCR, detection, layout analysis, and rule-based recovery pipeline than to an end-to-end VLM. The purpose of this comparison is to contextualize whether Task 1 mainly tests XML formatting or the harder problem of recovering editable diagram structure.

In our qualitative comparison, Paper2Any can produce visual outputs, but its decomposition is often coarse. It may treat the whole diagram as a single block or partition it into a small number of rectangular regions, producing slide-like patches rather than editable Draw.io objects. As a result, the recovered output lacks fine-grained nodes, edges, connectors, and topology

that can be directly manipulated in `mxGraph`. This supports the claim that Task 1 evaluates internal structure and editable semantic recovery, rather than merely converting an input into an XML-like file format.

We include the input diagrams, generated outputs, and visual comparisons for this baseline in the released benchmark repository together with the other supplementary experimental artifacts.

### A.1.2. PERFORMANCE TRADE-OFFS ON THE PARETO FRONT

Figure 8 visualizes the performance trade-offs between visual consistency (measured by SCS on Task 1) and editing fidelity (measured by XDRFR on Task 2) across frontier model families (GPT, Claude, Gemini). The scatter plot reveals several key insights: (1) **Performance correlation**: The positive correlation between SCS and XDRFR suggests that the underlying capabilities for structured understanding and code manipulation translate across tasks, validating that the code-mediated paradigm enables models to maintain both high visual fidelity and precise editability. Models from the Gemini, GPT, and Claude families tend to achieve strong performance on both dimensions simultaneously, indicating that strong visual code generation capabilities (Task 1) correlate with precise editing capabilities (Task 2). (2) **Model family characteristics**: Different model families may exhibit distinct positioning patterns, with some models potentially excelling in one dimension while maintaining competitive performance in the other. This analysis is crucial for understanding the transferability of capabilities between visual code generation and editing tasks, and demonstrates that the code-based representation provides a unified framework for both generation and manipulation of diagrammatic content.

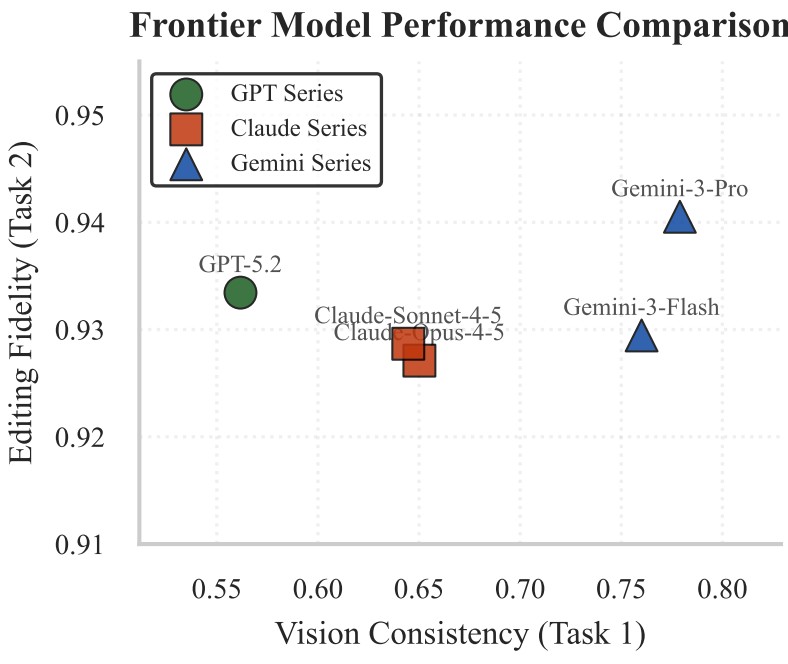

*Figure 8.* **Performance trade-offs**. This scatter plot compares the performance of frontier model families (GPT, Claude, Gemini) in terms of visual consistency (SCS for Task 1) and editing fidelity (XDRFR for Task 2).

### A.1.3. MODEL ROBUSTNESS ACROSS TASK DIFFICULTY LEVELS

Figure 9 presents a heatmap displaying CodeXQA accuracy scores for 11 evaluated models across Easy, Medium, and Hard difficulty levels on Task 1. The heatmap reveals several critical patterns: (1) **Top-tier performance**: The top 6-7 models (Gemini-3-pro, Gemini-3-flash, GPT-5.2, GLM-4.6V, Claude-sonnet-4-5, Claude-opus-4-5) consistently achieve high accuracy scores, generally above 0.90 for Easy and Medium tasks, and above 0.70 for Hard tasks. Notably, GPT-5.2 achieves the highest Easy score (0.951), while Gemini-3-pro leads on Medium (0.940) and Hard (0.895) tasks. (2) **Superior robustness of closed-source models**: These top-performing models demonstrate strong robustness with relatively small performance degradation as task difficulty increases. For instance, Gemini-3-pro's score only drops from 0.937 (Easy) to 0.895 (Hard), a decline of only 4.2 percentage points. (3) **Significant degradation for smaller models**: Models

like Qwen3-Omni-30B, Qwen3-VL-32B, and Qwen3-VL-8B show sharp performance declines with increasing difficulty. Qwen3-VL-8B performs particularly poorly, dropping from 0.164 (Easy) to 0.044 (Hard), representing a 73% relative decrease. (4) **Widening performance gap**: The difference between top-tier and lower-tier models becomes substantially wider as task difficulty increases, highlighting that complex diagrams require advanced reasoning capabilities currently concentrated in frontier models. This analysis validates the importance of difficulty stratification in our benchmark and reveals the current limitations of open-source alternatives in handling sophisticated visual code generation tasks.

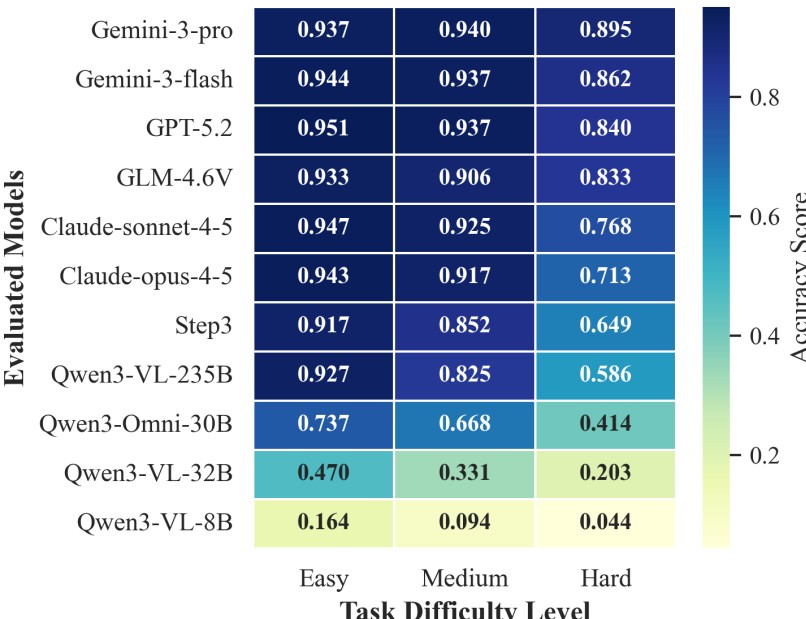

*Figure 9.* Model robustness across task difficulty levels. The heatmap shows CodeXQA accuracy scores of evaluated models on Easy, Medium, and Hard samples. Closed-source models demonstrate superior stability, while open-source and smaller-scale models show sharp performance degradation.

### A.1.4. XDRFR HUMAN CORRECTION AUDIT

To ensure the accuracy and reliability of the XDRFR evaluation metric, we conducted a manual audit of the automated evaluation results. We randomly sampled 100 evaluation samples (approximately 10% of the total 1,005 instructions) and manually reviewed the Yes/No answers generated by the automated evaluation model (gemini-3-pro-preview).

We selected 55 representative examples for detailed review. Each sample is uniquely identified by four dimensions: model name, domain, sample ID, and instruction. Among these 55 samples, only **one sample** (sample_0087) required manual correction of the automated evaluation answers, demonstrating the high reliability of our automated XDRFR evaluation pipeline. The corrected answers are highlighted in red in Table 11.

The sample requiring correction (sample_0087) is highlighted in red. This sample had two questions where the automated evaluation model (gemini-3-pro-preview) incorrectly answered "No" when the correct answer should have been "Yes". The extremely low correction rate (1 out of 55 samples, or 1.8%) demonstrates the high reliability of our automated XDRFR evaluation pipeline.

### A.1.5. DETAILED PERFORMANCE BREAKDOWN FOR TASK 2 ACROSS DIFFICULTY LEVELS

Table 12 presents the performance breakdown for Task 2 (instruction-to-patch editing) stratified by difficulty level. This table evaluates three key metrics: Execution Success Rate (ESR), Style Consistency Score (SCS), and XML Decomposed Requirements Following Ratio (XDRFR).

**Key Observations:**

- **Execution Success**: Nearly all models achieve near-perfect ESR ($\geq 0.99$) across difficulty levels, confirming that the

*Table 11.* XDRFR Human Correction Audit: 55 Representative Samples

| | | | | |
|---|---|---|---|---|
| sample_0087
`GLM-4.6`
*Move Denoise box...*
Q: 5, Manual Correction | sample_0008
`claude-opus-4-5`
*Change Start to red*
Q: 2, Correct | sample_0035
`gemini-3-flash`
*Delete Intermediate Step*
Q: 1, Correct | sample_0056
`gpt-5.2`
*Change arrow to blue...*
Q: 3, Correct | sample_0078
`claude-sonnet-4-5`
*Add Analysis node...*
Q: 3, Correct |
| sample_0092
`qwen3-vl-235b`
*Change central node...*
Q: 4, Correct | sample_0115
`glm-4.6`
*Move Navigation...*
Q: 2, Correct | sample_0134
`kimi-k2`
*Remove arrow...*
Q: 1, Correct | sample_0156
`minimax-m2`
*Change Process text...*
Q: 3, Correct | sample_0023
`qwen3-coder`
*Change User to yellow*
Q: 2, Correct |
| sample_0067
`deepseek-v3.2`
*Add Filter node...*
Q: 3, Correct | sample_0089
`claude-opus-4-5`
*Change Revenue border...*
Q: 2, Correct | sample_0101
`gemini-3-pro`
*Move Manager down...*
Q: 2, Correct | sample_0124
`gpt-5.2`
*Change Button shape...*
Q: 2, Correct | sample_0145
`qwen3-vl-32b`
*Delete connection...*
Q: 1, Correct |
| sample_0167
`glm-4.6`
*Change Payment text...*
Q: 2, Correct | sample_0189
`kimi-k2`
*Change arrow to double...*
Q: 2, Correct | sample_0203
`claude-sonnet-4-5`
*Change Main Topic size...*
Q: 2, Correct | sample_0221
`gemini-3-flash`
*Change Task 1 color...*
Q: 2, Correct | sample_0238
`qwen3-vl-235b`
*Change Neuron shape...*
Q: 2, Correct |
| sample_0195
`deepseek-v3.2`
*Change Customer bg...*
Q: 3, Correct | sample_0256
`gemini-3-pro`
*Change Sales text...*
Q: 2, Correct | sample_0267
`claude-opus-4-5`
*Delete Review node...*
Q: 2, Correct | sample_0278
`gpt-5.2`
*Change Login shape...*
Q: 2, Correct | sample_0289
`qwen3-vl-32b`
*Add Cache node...*
Q: 3, Correct |
| sample_0301
`kimi-k2`
*Change arrow to double...*
Q: 2, Correct | sample_0312
`glm-4.6`
*Change Main Topic size...*
Q: 2, Correct | sample_0323
`minimax-m2`
*Change SWOT border...*
Q: 2, Correct | sample_0334
`qwen3-coder`
*Move Phase 2 down...*
Q: 2, Correct | sample_0345
`claude-sonnet-4-5`
*Change Chart to yellow...*
Q: 2, Correct |
| sample_0356
`gemini-3-flash`
*Change Interface shape...*
Q: 2, Correct | sample_0367
`deepseek-v3.2`
*Change Feature A text...*
Q: 2, Correct | sample_0378
`qwen3-vl-235b`
*Change Layer shape...*
Q: 2, Correct | sample_0389
`gpt-5.2`
*Change Manager shape...*
Q: 2, Correct | sample_0401
`claude-opus-4-5`
*Change Header bg...*
Q: 2, Correct |
| sample_0412
`gemini-3-pro`
*Add Order table...*
Q: 2, Correct | sample_0423
`kimi-k2`
*Change Q1 Report shape...*
Q: 2, Correct | sample_0434
`glm-4.6`
*Change arrow to dashed...*
Q: 2, Correct | sample_0445
`qwen3-vl-32b`
*Delete Approval step...*
Q: 1, Correct | sample_0456
`minimax-m2`
*Change Branch 1 color...*
Q: 2, Correct |
| sample_0467
`claude-sonnet-4-5`
*Change arrow to blue...*
Q: 2, Correct | sample_0478
`qwen3-coder`
*Change Goal shape...*
Q: 2, Correct | sample_0489
`gemini-3-flash`
*Move Axis left...*
Q: 2, Correct | sample_0501
`deepseek-v3.2`
*Change Menu shape...*
Q: 2, Correct | sample_0512
`gpt-5.2`
*Change Task 5 border...*
Q: 2, Correct |
| sample_0523
`claude-opus-4-5`
*Change Activation shape...*
Q: 2, Correct | sample_0534
`qwen3-vl-235b`
*Add Utility class...*
Q: 2, Correct | sample_0545
`kimi-k2`
*Change Module B text...*
Q: 2, Correct | sample_0556
`glm-4.6`
*Change Team Lead shape...*
Q: 2, Correct | sample_0567
`minimax-m2`
*Delete connection...*
Q: 1, Correct |
| sample_0578
`claude-sonnet-4-5`
*Change Footer bg...*
Q: 2, Correct | sample_0589
`qwen3-vl-32b`
*Change Sub-branch 2...*
Q: 2, Correct | sample_0601
`gemini-3-pro`
*Add Product table...*
Q: 2, Correct | sample_0612
`deepseek-v3.2`
*Change Annual Report shape...*
Q: 2, Correct | sample_0623
`gpt-5.2`
*Move Verification...*
Q: 2, Correct |

*Table 12.* Task 2 performance breakdown across difficulty levels

| D. | C. | Model | ESR | SCS | XDRFR |
|---|---|---|---|---|---|
| Easy | Closed | Claude-4.5-Opus | **1.0000** | 0.8989 | 0.9254 |
| | | GPT-5.2 | **1.0000** | **0.9071** | 0.9428 |
| | | Gemini-3-Pro | **1.0000** | 0.9055 | **0.9466** |
| | | Gemini-3-Flash | **1.0000** | 0.8998 | 0.9344 |
| | | Claude-4.5-Sonnet | **1.0000** | 0.8832 | 0.9269 |
| | Open | Kimi-K2 | 0.9960 | 0.8886 | 0.9259 |
| | | GLM-4.6 | 0.9960 | 0.8779 | 0.9245 |
| | | Qwen3-VL-235B | 0.9980 | 0.8722 | **0.9302** |
| | | MiniMax-M2 | 0.9899 | 0.8693 | 0.9213 |
| | | Qwen3-VL-32B | **1.0000** | **0.9319** | 0.8841 |
| | | Qwen3-Coder | 0.9939 | 0.8907 | 0.9147 |
| | | DeepSeek-V3.2 | **1.0000** | 0.8899 | 0.8911 |
| Medium | Closed | Claude-4.5-Opus | 0.9971 | **0.9210** | 0.9256 |
| | | GPT-5.2 | 0.9971 | 0.9189 | 0.9214 |
| | | Gemini-3-Pro | 0.9985 | 0.9093 | **0.9379** |
| | | Gemini-3-Flash | **0.9985** | 0.9154 | 0.9308 |
| | | Claude-4.5-Sonnet | 0.9985 | 0.8952 | 0.9244 |
| | Open | Kimi-K2 | 0.9971 | 0.9138 | 0.9290 |
| | | GLM-4.6 | 0.9912 | 0.8995 | 0.9182 |
| | | Qwen3-VL-235B | 0.9956 | 0.9151 | **0.9311** |
| | | MiniMax-M2 | 0.9912 | 0.9114 | 0.9045 |
| | | Qwen3-VL-32B | **0.9985** | **0.9242** | 0.8807 |
| | | Qwen3-Coder | 0.9927 | 0.9127 | 0.9293 |
| | | DeepSeek-V3.2 | 0.9926 | 0.9167 | 0.8884 |
| Hard | Closed | Claude-4.5-Opus | **1.0000** | 0.9286 | 0.9336 |
| | | GPT-5.2 | 0.9951 | 0.9225 | **0.9526** |
| | | Gemini-3-Pro | **1.0000** | 0.9229 | 0.9389 |
| | | Gemini-3-Flash | **1.0000** | **0.9368** | 0.9188 |
| | | Claude-4.5-Sonnet | **1.0000** | 0.9139 | 0.9427 |
| | Open | Kimi-K2 | 0.9952 | 0.8901 | 0.9263 |
| | | GLM-4.6 | **1.0000** | 0.9389 | 0.9194 |
| | | Qwen3-VL-235B | 0.9951 | 0.9283 | 0.9330 |
| | | MiniMax-M2 | **1.0000** | 0.9160 | **0.9350** |
| | | Qwen3-VL-32B | 0.9952 | **0.9406** | 0.9097 |
| | | Qwen3-Coder | **1.0000** | 0.9186 | 0.9076 |
| | | DeepSeek-V3.2 | 0.9952 | 0.9206 | 0.9071 |

incremental modification format produces syntactically valid XML code reliably.

- **Style Preservation**: SCS scores remain consistently high (0.87-0.94) across all difficulty levels, indicating that models successfully maintain visual style and aesthetic quality even when performing complex edits. Closed-source models (especially Gemini-3-Flash and Claude-4.5-Opus) excel in style consistency.

- **Instruction Following**: XDRFR scores are uniformly high (0.88-0.95), significantly outperforming Task 1 performance, which validates the effectiveness of our proposed incremental modification strategy and the inherent controllability of the code-mediated paradigm. The decomposition-based evaluation enables precise verification of instruction compliance.

- **Difficulty Scaling**: Unlike Task 1, Task 2 performance shows remarkable stability across difficulty levels, with minimal degradation on Hard samples. This suggests that once diagrams are represented as code, editing operations become more tractable regardless of initial complexity.

- **Model Comparison**: Closed-source models maintain a slight edge, but open-source alternatives (particularly Qwen3-VL-235B, Kimi-K2, and GLM-4.6) demonstrate competitive performance, narrowing the gap compared to Task 1.

Bold values indicate the best performers in each category (closed-source/open-source) for each difficulty level and metric.

### A.1.6. PIXEL-BASED EDITING BASELINE FOR TASK 2

To empirically compare the Diagram-as-Code editing paradigm with a Diagram-as-Pixels alternative, we add a pixel-based editing baseline on a subset aligned with Task 2. The subset contains 32 diagrams, each paired with three editing instructions at the Easy, Medium, and Hard levels, resulting in 96 editing instructions in total. Both routes use the same English editing instructions. For the pixel-editing route, the input is the rendered PNG diagram and the editing instruction, and the output edited image is evaluated by human DRFR scoring over 335 decomposed questions. For the symbolic route, we follow the XML patch evaluation protocol used in the main paper and compute XDRFR on the same sampled instructions.

*Table 13.* Pixel-based editing baseline versus symbolic XML editing on a Task-2-aligned subset. DRFR is obtained from human evaluation of edited images, while XDRFR evaluates instruction compliance on the edited XML.

| Difficulty | Instructions | Questions | DRFR (Pixel) | XDRFR (XML) |
|---|---|---|---|---|
| Overall | 96 | 335 | 0.821 | 0.940 |
| Easy | 32 | 50 | 0.960 | 0.983 |
| Medium | 32 | 97 | 0.856 | 0.969 |
| Hard | 32 | 188 | 0.649 | 0.867 |

The results show that the pixel-based baseline is consistently weaker than the symbolic XML editing route, both overall and across all difficulty levels. The gap is most pronounced on Hard instructions, where the pixel baseline drops to 0.649 DRFR, while XML-based editing maintains 0.867 XDRFR. Qualitatively, pixel editing often introduces blurred or incorrectly modified text, unintentionally deletes or changes irrelevant elements, and struggles to preserve connectors and topology. Even when the requested edit is partially applied, geometric positions, font sizes, and colors are often difficult to control precisely. Fine-grained local edits, such as slightly enlarging an object or moving it a little to the left, are especially unreliable in pixel space. In contrast, the XML-based route applies localized patches, re-renders through the same evaluation pipeline, and preserves the option of opening the result in Draw.io for further structured refinement. This makes code-mediated editing better aligned with professional diagram-editing workflows.

## B. Dataset Details

### B.1. Dataset Taxonomy

The VCG-Bench dataset is designed to cover a wide range of structured diagrams used in professional settings. We categorize the 1,449 examples into 6 major domains (L1) and 15 sub-domains (L2). This section provides a detailed breakdown of each category.

**Academic Domain**    This domain focuses on scientific research, algorithmic logic, and data representation, which are critical for academic communication.

- `System Architecture & Flow`: This category includes algorithmic logic diagrams, technical architecture blueprints, and model training pipelines. These diagrams typically involve complex node connections and directed edges representing data or control flow.

- `Neural Networks`: Specific to deep learning, these diagrams feature neurons, layer stacking, convolution kernels, and feature maps. They often require precise alignment and repetitive structures.

- `Data Visualization`: This includes tensor transformations, experimental histograms, scatter plots, and coordinate axes, emphasizing precise quantitative representation.

**Software Domain**    This domain covers standardized modeling languages essential in software engineering.

- `UML Class Diagrams`: These are typical three-tier rectangles showing Class, Property, and Method. They are fundamental for object-oriented design and require strict adherence to UML standards.

- `Sequence Diagrams`: These logic diagrams use vertical dashed lines (lifelines) and horizontal interaction arrows to depict the sequence of messages between objects.

- `Database ER Diagrams`: Entity-relationship diagrams that illustrate table structures and primary/foreign key associations, crucial for database design.

**Business Domain**    This domain emphasizes business planning, corporate operations, and product layout.

- `Product Architecture`: These diagrams illustrate product functional modules, middle-office components, or platform-level stacks.

- `Performance Reports`: PPT-style summaries featuring KPI figures and titled reporting slides, often used in business presentations.

- `Business Strategy Models`: This includes classic management models such as SWOT analysis, PDCA cycles, and Porter's Five Forces.

**Management Domain**    This domain is used for organizational collaboration, task tracking, and process standardization.

- `Gantt Progress`: Charts used for project schedule tracking.

- `Organizational Hierarchy`: Diagrams depicting the structure of an organization.

- `Project Flowcharts`: Standard Operating Procedures (SOPs) and workflow diagrams.

**UIUX Domain**    This domain involves interaction design prototypes for digital products.

- `Web Prototypes`: Wireframes and low-fidelity prototypes for websites.

- `Mobile Wireframes`: Interface designs tailored for mobile devices with specific aspect ratios.

**General Domain**    This domain includes general logical thinking tools.

- `Mind Maps`: Characterized by radiant or tree-like logical connection diagrams emanating from a central topic, used for brainstorming and organizing ideas.

## B.2. Data Source Details

To ensure transparency and reproducibility, we detail our four primary data sources:

1. **Open-Source Template Repositories (40%)**: Sourced from public GitHub repositories hosting '.drawio' or '.xml' templates (e.g., 'jgraph/drawio-diagrams', architecture-templates).

2. **Open Access Academic Papers (30%)**: Extracted from arXiv sources under CC-BY licenses, focusing on CS/AI system architectures.

3. **Anonymized Corporate Diagrams (20%)**: Internal datasets from verified industry partners. All text entities were anonymized (e.g., "Feature A" instead of specific product names) and PII was scrubbed using regex and NER pipelines.

4. **Permissively Licensed Web Diagrams (10%)**: Crawled from technical blogs and documentation sites explicitly marked as CC-BY or public domain.

## B.3. Data Synthesis Pipeline

The VCG-Bench dataset is constructed using a two-stage synthesis pipeline. Gemini-3-Pro is used to generate intermediate structured captions, while candidate `mxGraph` XML files are synthesized with multiple VLM candidates and retained through executability, rendering-consistency, and human-verification filters. This data-construction process is separate from the formal benchmark evaluation, where the evaluated models are compared on the same released Task 1 and Task 2 inputs.

**Stage 1: Image-to-`mxGraph` XML (Task 1)**

1. **Source Collection**: Images gathered from public repositories (Design2Code, UMLModels) and web crawls.

2. **Screening**: Filtered for diagrammatic clarity and content.

3. **Description Generation**: Intermediate structured JSON describing 'components', 'spatial_layout', and 'overview'.

4. **Code Generation (Task 1)**: Converting the description and the original image into `mxGraphModel` mxGraph XML.

5. **Verification**: Rendering the `mxGraph` XML and filtering based on visual similarity.

**Stage 2: Instruction Synthesis (Task 2)**

1. **Base Selection**: High-quality `mxGraph` XMLs from Stage 1 are selected as ground truth.

2. **Instruction Generation**: Creating instructions with varying difficulty levels (Easy, Medium, Hard).

3. **Execution & GT Generation**: Applying edits via code modification to generate the "After" `mxGraph` XML state.

4. **Atomic Operations**: Edits are composed of 14 atomic operations (e.g., `add_node`, `change_color`, `reroute_edge`).

## B.4. Dataset Statistics and Difficulty

**Task 1 Difficulty (`mxGraph` XML Token Count)**:

- **Easy** (< 8,645 tokens): 33.0% (478 images)

- **Medium** (8,645 - 14,000 tokens): 49.8% (721 images)

- **Hard** (> 14,000 tokens): 17.3% (250 images)

*Statistics*: Mean 11,024, Median 10,009.

**Task 2 Difficulty (Atomic Operations)**:

- **Easy**: 1-2 atomic operations.

- **Medium**: 3-4 atomic operations.

- **Hard**: 5-7 atomic operations.

Total: 1,005 editing instructions derived from 335 unique base diagrams.

### B.5. Dataset and Licensing

This appendix provides additional details on the dataset composition, licensing, and curation policies supporting VCG-Bench.

- **Sources**: publicly available diagrams from research papers and open repositories.

- **Licensing**: diagrams are included under compliant open licenses; we respect attribution and redistribution terms.

- **Content**: non-template, compositional layouts featuring mixed shapes, connectors, and textual labels.

- **Diversity**: balanced styles (palettes, line types, arrowheads), densities, and connector usage to avoid narrow template bias.

#### Sources and Diversity

We specifically select diagrams that emphasize conceptual/system sketches and mixed-shape schemas over rigid templates. Diversity targets include:

- **Shape categories**: rectangles, rounded boxes, ellipses, diamonds, custom shapes.

- **Connectors**: directed edges with single/multiple waypoints, orthogonal/curved lines, different arrowheads.

- **Text**: node labels, inline annotations, legends, captions.

## C. Additional Information

### C.1. Annotation Schema Details

Each image is annotated at element-level granularity for robust structured encoding.

- Nodes: shape type, bounding box, size, style (fill/line), text content.

- Edges: source/target attachments, waypoints, line style, arrowheads, labels (if any).

- Groups/Layers: hierarchical organization to preserve edit semantics and complex layouts.

- Relations: attachment, grouping, and layer membership enabling consistent edits (move/resize without breaking connectors).

### C.2. Ethical Considerations and Release

We comply with licensing, attribution, and privacy norms. Sensitive or proprietary diagrams are excluded. Release artifacts include annotated images and valid, editable XML to enable reproducible evaluation.

### C.3. Discussion

Beyond evaluation, VCG-Bench serves as a scalable data engine capable of generating high-quality, verified visual-code pairs, addressing the chronic scarcity of structured diagrammatic data for VLM training.

Figure 10 illustrates the data flywheel mechanism enabled by our pipeline. The flywheel operates through a self-reinforcing cycle with four interconnected components: (1) **VCG Pipeline**: The pipeline processes raw input through three sequential

stages: evaluation (assessment and validation), filtering (selection and refinement), and enhancement (improvement and augmentation), ultimately producing high-quality training data. The pipeline receives input from the Data Flywheel, creating an iterative improvement mechanism. (2) **Data Flywheel**: A circular mechanism with four colored segments (orange, red, teal, green) arranged in a continuous clockwise cycle, driving the pipeline's input and receiving feedback to refine the data generation process. (3) **Model Training Applications**: The high-quality training data serves three distinct model types: **VLM (Vision2Text)** for vision-to-text conversion, **LLM (XML2XML)** for XML-to-XML processing, and **Diffusion (Vision2Vision)** for vision-to-vision generation. Each model type benefits from the structured visual-code pairs with precise annotations. (4) **Feedback Loop**: Human feedback (represented by stylized human figures with stars) flows from the training data back to the Data Flywheel, completing the self-improving cycle. This feedback mechanism allows the system to incorporate human judgment and domain expertise, refining both data quality and model performance. The flywheel addresses a fundamental bottleneck in visual code generation research: the lack of large-scale, high-quality training data with precise structural annotations. By providing both evaluation benchmarks and scalable data generation capabilities with enhanced controllability and explainability, VCG-Bench contributes to advancing the field beyond mere performance measurement, enabling a continuous improvement cycle for visual code generation systems. As frontier VLMs depend on efficient long-context inference to handle the verbose structured outputs inherent to `mxGraph` XML (Liu et al., 2025), the verifiable visual-code pairs produced by VCG-Bench offer a targeted training signal for specializing these large models in diagram-centric tasks.

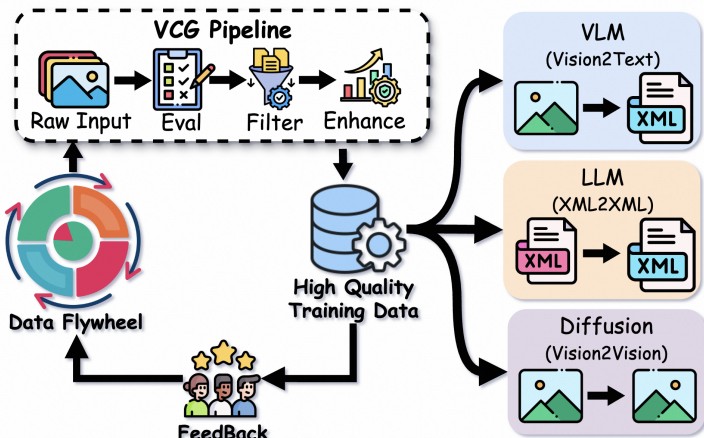

*Figure 10.* VCG-Bench Pipeline as a crucial Data Flywheel, delivering high-quality training data for VLM, LLM, and Diffusion models with enhanced controllability and explainability.

## D. Technical Specifications

### D.1. mxGraph XML Schema Overview

Ground-truth `mxGraph` XML follows the mxGraph structure and loads without repair in standard editors. A minimal illustrative snippet:

```
1  <mxGraph>
2    <root>
3      <mxCell id="0"/>
4      <mxCell id="1" parent="0"/>
5      <mxCell id="2" value="Process" style="rounded=1;whiteSpace=wrap;"
6            vertex="1" parent="1">
7        <mxGeometry x="60" y="40" width="120" height="60" as="geometry"/>
8      </mxCell>
9      <mxCell id="3" edge="1" parent="1" source="2" target="4"
10           style="endArrow=block;orthogonalLoop=1;">
11       <mxGeometry relative="1" as="geometry">
12         <mxPoint x="120" y="100" as="targetPoint"/>
13       </mxGeometry>
14     </mxCell>
15     <mxCell id="4" value="Output" vertex="1" parent="1">
```

```
16        <mxGeometry x="260" y="40" width="120" height="60" as="geometry"/>
17      </mxCell>
18    </root>
19  </mxGraph>
```

This structure encodes nodes, connectors, geometry, and styles for accurate reconstruction and editability.

### D.2. Comparative Analysis of Visual Representation Formats

To contextualize the significance of `mxGraph`, we provide a deeper comparison with `SVG`, `HTML`, and `PowerPoint` (PPT) on three critical dimensions:

- **Semantic Depth vs. Graphical Primitives**: Unlike `SVG`, which primarily defines low-level vector paths and shapes, `mxGraph`'s `mxGraph` XML schema explicitly encodes semantic entities (e.g., nodes, actors) and their logical connections (e.g., edges, directional flows). This abstraction allows models to grasp the underlying system topology rather than just pixel positions.

- **Structured Editability vs. Layout Rigidity**: `HTML/CSS` is optimized for document flow and responsive layouts but struggles with the arbitrary spatial relationships inherent in diagrams. In contrast, `mxGraph` offers a flexible canvas that supports precise, absolute positioning while maintaining the structural integrity required for diagrammatic logic, enabling robust round-trip editing unavailable in rigid DOM structures.

- **Open Interpretability vs. Proprietary Formats**: While `PowerPoint` (PPT) is widely used, its proprietary and complex file structure hinders programmatic generation and fine-grained manipulation. `mxGraph` utilizes an open, human-readable `mxGraph` XML format that bridges the gap between visual design and code-based reasoning, making it an ideal standard for evaluating structured generation capabilities in AI agents.

---

**Format Specification: `mxGraphModel` XML (Draw.io)**

The Draw.io diagram format is based on an `mxGraphModel` XML document. In this benchmark, we assume the following minimal structural requirements for a document to be considered valid:

- **Document structure**: The `mxGraph` XML document contains an `<mxGraphModel>` element with a `<root>` child. The `<root>` element includes (at minimum) `<mxCell id="0"/>` and `<mxCell id="1" parent="0"/>`.

- **Cell metadata**: Each diagram element is encoded as an `mxCell` with attributes including `id` (unique identifier), `parent` (hierarchical containment), `value` (label text, if present), and `style` (visual encoding).

- **Geometric and connectivity information**: Node placement is specified via an `<mxGeometry>` element with `x`, `y`, `width`, and `height` attributes. Edge connectivity is specified using the `source` and `target` attributes on the corresponding `mxCell`.

For brevity, we refer to this representation as **`mxGraph` XML** throughout.

---

## E. Prompt Templates and Instructions

We provide the core prompt templates used in our data synthesis and evaluation pipelines. These prompts utilize `gemini-3-pro-preview` for high-quality generation and reasoning.

### E.1. Task 1: Diagram Generation Prompts

E.1.1. FLOWCHART CLASSIFICATION PROMPT

**Purpose**: Determine whether an image is a flowchart/architecture diagram that can be faithfully reconstructed using Draw.io/XML.

```
1    Analyze the provided image and determine if it is a flowchart or architecture diagram suitable for Draw.io/XML
          reconstruction.
2
```

```
3    **Criteria**: The image should contain discrete nodes/shapes connected by arrows or lines, representing a
         system, process, or workflow. Exclude data visualizations (charts, plots), screenshots, photographs, and
         images with figurative characters.
4
5    **Output Format (JSON)**:
6    {
7      "is_candidate": true/false,
8      "score": 0.0-1.0,
9      "diagram_type": "flowchart|architecture|network|dataflow|other|null",
10     "reason": "Brief explanation"
11   }
```

*Listing 1.* Prompt for Flowchart Classification

### E.1.2. DIAGRAM DESCRIPTION GENERATION PROMPT

**Purpose**: Generate a detailed structured description of the diagram (JSON format).

```
1    Analyze the provided diagram image and generate a structured JSON description.
2
3    **Context**: {context_text} (optional, use if provided)
4
5    **Requirements**:
6    - Identify all components (nodes, shapes, text labels) with their styles and positions
7    - Trace all connections/arrows with their properties (from/to, color, style, routing)
8    - Describe spatial layout and relationships
9    - Extract color palette and visual styles
10
11   **Output Format**: Output ONLY a JSON object (no Markdown) with fields:
12   - "image_type": string
13   - "overview": string
14   - "components": [{"name": string, "count": number, "description": string}]
15   - "spatial_layout": {"primary_layout": string, "relative_positions": [string], "alignment": [string]}
16   - "flow_and_topology": {"primary_flow": string, "structures": [string]}
17   - "arrows": {"summary": {...}, "details": [{...}]} (optional)
18   - "component_styles": [{...}] (optional)
19   - "color_palette": {...} (optional)
20   - Other optional fields as needed
21
22   Use the image as ground truth for all descriptions.
```

*Listing 2.* Prompt for Diagram Description Generation

### E.1.3. XML GENERATION PROMPT

**Purpose**: Generate Draw.io `mxGraph` XML code based on the image and description.

```
1    You are a Draw.io XML generation expert. Generate a complete, valid Draw.io XML file based on the [JSON
         Description Draft] and [Original Image].
2
3    **Inputs**:
4    - [JSON Description Draft]: {json_description}
5    - [Original Image]: (use as ground truth for verification)
6
7    **XML Requirements**:
8    1. Output ONLY XML content (no Markdown fences, no explanations)
9    2. Start with `<mxfile>` and end with `</mxfile>`
10   3. `<mxGraphModel>` must include `grid="1" gridSize="10" guides="1" page="1"`
11   4. All elements must be `<mxCell>` with:
12     - Unique `id` (globally unique, add suffixes if needed)
13     - `value`, `style`, `parent` attributes
14     - `<mxGeometry ... as="geometry"/>` child
15   5. Edge cells (`edge="1"`) must NOT be self-closing; must contain `<mxGeometry relative="1" as="geometry" />`
16   6. XML escaping: Use `<`, `>`, `&` in attribute values
17   7. No external images (no `shape=image` with URLs)
18
19   **Generation Rules**:
20   1. Use [Original Image] as ground truth; [JSON Description Draft] as reference
21   2. Layout: All coordinates must be multiples of 10. Center the diagram on canvas (don't stack in top-left
         corner)
22   3. Components: Create nodes from `components` and `component_styles`. Use styles from JSON when available
23   4. Arrows: Create exactly one `<mxCell edge="1">` for each item in `arrows.details`:
24     - Use `style` field (solid/dashed)
25     - Set `startArrow`/`endArrow` based on `heads` field (double arrow if `is_bidirectional: true`)
26     - Use `routing` field: add `edgeStyle=orthogonalEdgeStyle;` for orthogonal, use `<Array as="points">` for
         curved
```

```
27        – Set `exitX`, `exitY`, `entryX`, `entryY` based on `connection_ports` (e.g., "middle_right" -> `exitX=1;
             exitY=0.5;`)
28     5. Z-Order: Create background elements (with `is_background: true`) first, then foreground elements
29     6. Complex structures: For `relationships_extended` with `type: "describes_subgraph"` or `"describes_structure
          "`, rebuild the structure using `<mxCell>` primitives based on the `note` description
30
31     **Validation**: Ensure all `id`s are unique, all edges have proper source/target references, and XML is well-
          formed.
```

*Listing 3.* Prompt for XML Generation

## E.2. Task 2: Diagram Editing Prompts

### E.2.1. INSTRUCTION GENERATION PROMPT

**Purpose**: Generate 3 editing instructions of different difficulty levels (Easy/Medium/Hard) for a given diagram.

```
1     You are a professional diagram editing instruction generation expert. Your task is to generate **3
          instructions of different difficulty levels** for the given diagram.
2
3     ## Input Information
4
5     **\texttt{mxGraph} XML Structure**:
6     ```xml
7     {xml_content}
8     ```
9
10    **Rendered Image**: Please examine the provided rendered image to understand the visual structure of the
          diagram.
11
12    ---
13
14    ## Atomic Operations Definition
15
16    The following are **14 predefined atomic operations**, each being the most basic, indivisible operation unit
          in diagram editing:
17
18    ### Category 1: Node Attribute Modification (4 operations)
19    1. **Modify Node Color**: Change the fill color or border color of a node
20    2. **Modify Node Shape**: Change the shape type of a node (rectangle, circle, diamond, etc.)
21    3. **Modify Node Size**: Change the width or height of a node
22    4. **Modify Node Text**: Change the text content displayed inside a node
23
24    ### Category 2: Node Structure Operations (3 operations)
25    5. **Delete Node**: Delete a node (without handling connections)
26    6. **Add Node**: Add a new node at a specified location
27    7. **Move Node**: Change the position of a node
28
29    ### Category 3: Connection Line Attribute Modification (3 operations)
30    8. **Modify Connection Color**: Change the color of a connection line
31    9. **Modify Connection Style**: Change the style of a connection line (solid, dashed, thickness, etc.)
32    10. **Modify Connection Arrow**: Change the arrow style of a connection line
33
34    ### Category 4: Connection Line Structure Operations (4 operations)
35    11. **Delete Connection**: Delete a connection line
36    12. **Add Connection**: Add a connection line between two nodes
37    13. **Redirect Connection**: Change the start or end point of a connection line
38    14. **Update Connection Path**: Update the connection line path when nodes move
39
40    ---
41
42    ## Difficulty Requirements (Strictly Follow)
43
44    You need to generate 3 instructions of different difficulty levels, where difficulty is **completely
          determined by the number of atomic operations**:
45
46    ### Easy Difficulty (1-2 atomic operations)
47    - **Requirement**: The instruction must contain **1-2** of the above 14 atomic operations
48
49    ### Medium Difficulty (3-4 atomic operations)
50    - **Requirement**: The instruction must contain **3-4** of the above 14 atomic operations
51
52    ### Hard Difficulty (5-7 atomic operations)
53    - **Requirement**: The instruction must contain **5-7** of the above 14 atomic operations
54
55    ---
56
57    ## Natural Language Instruction Requirements (Core Requirement)
```

```
58
59    **Most Important Principle**: Instructions must be based on the **rendered image** that users see, using
          natural language, not based on \texttt{mxGraph} XML code or technical identifiers.
60
61    ### [RECOMMENDED] Recommended Description Methods (Prioritized):
62
63    **Highest Priority**: Directly use the text content displayed on nodes
64
65    1. **Node Text Content Description (Most natural and accurate)**:
66       - "Change the 'Start' node to red"
67       - "Delete the 'Data Processing' node"
68
69    2. **Semantic Description** (when nodes don't have clear text but semantics are clear):
70       - "Start node", "End node"
71
72    3. **Position Description** (when nodes have no text or text is unclear):
73       - "The topmost node", "The leftmost rectangle", "The middle circle"
74
75    4. **Visual Feature Description** (as supplementary positioning):
76       - "The largest rectangle", "The red node"
77
78    5. **Relative Position Description**:
79       - "The arrow between 'Start' and 'End'"
80
81    ### [PROHIBITED] Strictly Prohibited Description Methods:
82
83    1. **Technical Identifiers** (Prohibited):
84       - [X] "Node with id 'node_1'"
85       - [X] "edge_5"
86
87    2. **Coordinate Description** (Prohibited):
88       - [X] "Node at coordinates (200, 300)"
89
90    3. **XML Attribute Reference** (Prohibited):
91       - [X] "Node with shape='rectangle'"
92
93    ---
94
95    ## Output Format (Strict JSON)
96
97    Please output a JSON object containing 3 instructions in the following format:
98
99    ```json
100   {
101     "instructions": [
102       {
103         "difficulty": "Easy",
104         "instruction": "Change the 'Start' node to red",
105         "atomic_operations": [
106           {
107             "operation_id": 1,
108             "operation_type": "Modify Node Color",
109             "description": "Change the color of the 'Start' node to red"
110           }
111         ],
112         "atomic_operation_count": 1,
113         "target_elements": [
114           {
115             "type": "node",
116             "visual_description": "The 'Start' node"
117           }
118         ],
119         "executable": true
120       },
121       {
122         "difficulty": "Medium",
123         ...
124       },
125       {
126         "difficulty": "Hard",
127         ...
128       }
129     ]
130   }
131   ```
```

*Listing 4.* Prompt for Instruction Generation

### E.2.2. INSTRUCTION DECOMPOSITION PROMPT (XDRFR)

**Purpose**: Decompose a natural language editing instruction into a series of verifiable Yes/No questions.

```
1   Decompose the following diagram modification instruction into a series of Yes/No questions. These questions
        will be used to verify whether the instruction has been correctly followed.
2   These questions will be verified based on \texttt{mxGraph} XML code, not rendered images.
3
4   **Important Note**: The instruction uses natural language descriptions (based on rendered images), but
        questions need to be converted to \texttt{mxGraph} XML code-level verification.
5   For example, if the instruction says "Change the 'Start' node to red", the question should be converted to "
        Has the fillColor attribute of the node with value 'Start' been changed to red?"
6
7   Instruction: "{instruction}"
8
9   **Requirements:**
10
11  **1. Target only the instruction itself**
12  - Carefully analyze the specific modification content mentioned in the instruction
13  - Decompose each specific requirement in the instruction into independent Yes/No questions
14  - Do not add content or checks not mentioned in the instruction
15
16  **2. Question format requirements**
17  - Each question should be answerable with "Yes" or "No"
18  - Questions should be verifiable by comparing the original \texttt{mxGraph} XML code and the modified \texttt{
        mxGraph} XML code
19  - Questions should be clear and unambiguous
20  - Questions should directly correspond to the specific modification requirements in the instruction
21  - Questions should focus on \texttt{mxGraph} XML code-level modifications (attribute values, element existence,
             structural changes, etc.)
22  - **Key**: Convert natural language descriptions to \texttt{mxGraph} XML code-level verification methods
23
24  **3. Natural language to \texttt{mxGraph} XML mapping rules**
25  - **Node text description** (e.g., "Change the 'Start' node...") -> Find nodes in \texttt{mxGraph} XML where
        the 'value' attribute contains that text
26  - **Position description** (e.g., "the topmost node") -> Need to combine `mxGeometry` coordinate information
        to locate (smallest y coordinate)
27  - **Semantic description** (e.g., "start node") -> Find nodes where the `value` attribute contains related
        semantics
28  - **Color description** (e.g., "change to red") -> Verify `fillColor` or `strokeColor` in the `style`
        attribute (support color names or hex codes)
29  - **Shape description** (e.g., "change to circle") -> Verify `shape` or related attributes in the `style`
        attribute
30  - **Connection description** (e.g., "arrow connecting 'Input' and 'Output'") -> Find nodes where `source` and `
        target` attributes correspond (match by value attribute)
31
32  **Output Format (JSON):**
33  {
34      "decomposed_questions": [
35          "Has the fillColor attribute of the node with value 'Start' been changed to red (#FF0000 or red)?",
36          "Does the node with value 'Start' still exist in the modified \texttt{mxGraph} XML code?"
37      ]
38  }
```

*Listing 5.* Prompt for XDRFR Instruction Decomposition

### E.2.3. XML EDITING PROMPT

**Purpose**: Modify `mxGraph` XML code according to the instruction (incremental format).

```
1   You are a professional Draw.io XML editing expert. Your task is to edit the given XML according to the
        modification instruction.
2
3   ## Original \texttt{mxGraph} XML
4   ```xml
5   {original_xml}
6   ```
7
8   ## Modification Instruction
9   {instruction}
10
11  ## Requirements
12  1. **Only output modified parts**: To save tokens, you only need to output the modified \texttt{mxGraph} XML
        fragments, not the complete \texttt{mxGraph} XML
13  2. **Incremental format**: Output JSON format containing all places that need modification (because there may
        be multiple places to modify)
14  3. **Format requirements**:
15     - Each modification contains two fields:
```

```
16          – 'original_fragment': Original \texttt{mxGraph} XML fragment (the part to be replaced)
17          – 'modified_fragment': Modified \texttt{mxGraph} XML fragment (the replacement content)
18        – If there are multiple modifications, output multiple modification objects
19     4. **Fragment requirements**:
20        – 'original_fragment' must be a complete fragment from the original \texttt{mxGraph} XML (can be a complete
              element, such as '<mxCell>...</mxCell>')
21        – 'modified_fragment' is the corresponding modified fragment
22        – Fragments must be precise enough to uniquely match the position in the original \texttt{mxGraph} XML
23     5. **Maintain correct \texttt{mxGraph} XML format**: Modified fragments must maintain correct and parseable \
           texttt{mxGraph} XML format
24
25     ## Output Format (Strict JSON)
26
27     Please output a JSON object in the following format:
28
29     ```json
30     {
31       "changes": [
32         {
33           "original_fragment": "<mxCell id='node_1' value='Old Text' ...>...</mxCell>",
34           "modified_fragment": "<mxCell id='node_1' value='New Text' ...>...</mxCell>"
35         },
36         {
37           "original_fragment": "<mxCell id='edge_1' ...>...</mxCell>",
38           "modified_fragment": "<mxCell id='edge_1' ...>...</mxCell>"
39         }
40       ]
41     }
42     ```
43
44     **Important Notes**:
45     – If the instruction involves only one modification, the 'changes' array contains only one element
46     – If the instruction involves multiple modifications (e.g., "delete node A and node B"), the 'changes' array
           contains multiple elements
47     – 'original_fragment' must be a complete and precise fragment from the original \texttt{mxGraph} XML
48     – For deletion operations, 'modified_fragment' is an empty string '""'
49     – Only output JSON, do not include any explanatory text or Markdown code block markers
```

*Listing 6.* Prompt for XML Editing

## E.3. Evaluation-Related Prompts

### E.3.1. SCS (STYLE CONSISTENCY SCORE)

**SCS Prompt (Task 1)**  **Purpose**: Evaluate style consistency between the generated diagram and the original diagram (Task 1).

```
1     You are a professional diagram design reviewer. Please compare the "original diagram" and "generated diagram"
          and conduct a systematic evaluation following these steps:
2
3     Original diagram:
4     [Image: {original_path}]
5
6     Generated diagram:
7     [Image: {generated_path}]
8
9     **Evaluation Steps:**
10
11    **Step 1: Attribute Extraction**
12    Please carefully analyze the original diagram and extract the following key attributes:
13    – Color scheme: List the main colors' hexadecimal values or color family names (e.g., "blue family", "warm
          tones")
14    – Font style: Font weight (thin/medium/bold), size (small/medium/large), font family (if any)
15    – Layout structure: Overall structure type (tree/ring/linear/grid/free layout)
16    – Visual elements: Node shapes (circle/rectangle/diamond, etc.), line styles (solid/dashed/thickness)
17
18    **Step 2: Difference Comparison**
19    Check the deviation of the above attributes in the generated diagram one by one:
20    – Is the color scheme consistent? If there are differences, describe the specific differences
21    – Does the font style match?
22    – Is the layout structure the same? Are element positions and spacing relationships consistent?
23    – Are visual element styles consistent?
24
25    **Step 3: Dimension Scoring (0-10 scale)**
26    Please score the following three dimensions separately (0-10 points, keep one decimal place):
27    1. Visual style consistency (color, line thickness, node style): ___
28    2. Layout structure consistency (element positions, spacing, spatial relationships): ___
```

```
29      3. Aesthetic quality (alignment, visual balance, overall beauty): ___
30
31      **Step 4: Final Score Calculation**
32      – Calculate the average of the three dimension scores
33      – Divide the average by 10 to normalize to 0–1
34      – Example: Dimension scores [8.5, 7.0, 9.0], average = 8.17, final score = 0.817
35
36      **Output Format (JSON):**
37      {
38         "analysis": {
39            "original_attributes": {
40               "color_scheme": "...",
41               "font_style": "...",
42               "layout_structure": "...",
43               "visual_elements": "..."
44            },
45            "differences": [
46               "Difference description 1",
47               "Difference description 2"
48            ],
49            "dimension_scores": {
50               "visual_style_consistency": 8.5,
51               "layout_consistency": 7.0,
52               "aesthetic_quality": 9.0
53            },
54            "average_score": 8.17
55         },
56         "final_score": 0.817
57      }
58
59      **Important Notes:**
60      – Please strictly follow the scoring criteria and avoid directly giving high scores (0.85+)
61      – You must complete attribute extraction and difference comparison before giving dimension scores
62      – If the generated diagram has obvious visual errors (such as element overlap, text misalignment), points
            should be deducted in the corresponding dimension
63      – The final score must be calculated based on the average of dimension scores, not estimated directly
```

*Listing 7.* SCS Prompt for Task 1

**SCS Prompt (Task 2)** **Purpose**: Evaluate style and aesthetic consistency between the modified diagram and the original diagram (Task 2; only style and aesthetics are evaluated).

```
1       You are a professional diagram design reviewer. Please compare the "Original Diagram (Gemini generated)" and
            the "Modified Diagram (Model generated)" to evaluate whether the modified diagram maintains the original
            **style and aesthetic**.
2
3       **Important Notes**:
4       – The modified diagram will definitely have changes (elements may be deleted, added, modified), which is **
            normal** and should not result in score deduction
5       – **Only evaluate style consistency and aesthetic consistency**, do not evaluate element completeness, text
            consistency, or other content consistency
6       – We focus on: whether the modified diagram overall still conforms to the original diagram's style and
            aesthetic standards
7
8       Original Diagram (Gemini generated):
9       [Image: {gemini_rendered_path}]
10
11      Modified Diagram (Model generated):
12      [Image: {modified_rendered_path}]
13
14      **Evaluation Steps:**
15
16      **Step 1: Style Attribute Extraction**
17      Please carefully analyze the original diagram and extract the following **style and aesthetic** related key
            attributes:
18      – **Color Scheme**: Main color families, tones, saturation characteristics (e.g., "blue family", "warm tones",
            "high saturation")
19      – **Font Style**: Overall style characteristics of font weight, size, font family
20      – **Visual Element Style**: Uniformity of node shapes, uniformity of line styles, overall visual style
21      – **Aesthetic Features**: Overall layout balance, alignment methods, visual hierarchy, space utilization
22
23      **Step 2: Style and Aesthetic Consistency Evaluation**
24      Evaluate the modified diagram's performance in the following aspects:
25      – **Color Style Consistency**: Do unmodified parts maintain the original color style? Is the overall color
            scheme harmonious?
26      – **Visual Style Consistency**: Do unmodified parts maintain the original visual style (node shapes, line
            styles, etc.)?
27      – **Aesthetic Quality**: Is the modified diagram overall still beautiful, balanced, and professional? Are
            there obvious visual errors (element overlaps, text misalignment, layout chaos)?
```

```
28
29    **Important Evaluation Principles**:
30    - [YES] **Should evaluate**: Style consistency of unmodified parts, overall aesthetic quality, visual errors
31    - [NO] **Should NOT evaluate**: Whether element counts are consistent, whether text content is consistent,
           whether elements are complete (these changes are normal)
32
33    **Step 3: Dimension Scoring (0-10 scale)**
34    Please score the following two dimensions separately (0-10 points, keep one decimal place):
35    1. **Style Consistency** (color style, visual element style, overall style characteristics): ___
36       - Evaluate whether unmodified parts maintain the original style characteristics
37       - Evaluate whether the overall style is coordinated and unified
38    2. **Aesthetic Quality** (visual balance, alignment, overall beauty, presence of obvious visual errors): ___
39       - Evaluate whether the modified diagram is still beautiful and professional
40       - If there are obvious visual errors (element overlaps, text misalignment, layout chaos), deduct points
41
42    **Step 4: Final Score Calculation**
43    - Calculate the average of the two dimension scores
44    - Divide the average by 10 to normalize to 0-1
45    - Example: Dimension scores [8.5, 9.0], average = 8.75, final score = 0.875
46
47    **Output Format (JSON):**
48    {
49        "analysis": {
50           "original_attributes": {
51              "color_scheme": "...",
52              "font_style": "...",
53              "visual_element_style": "...",
54              "aesthetic_features": "..."
55           },
56           "differences": [
57              "Difference description 1",
58              "Difference description 2"
59           ],
60           "dimension_scores": {
61              "style_consistency": 8.5,
62              "aesthetic_quality": 9.0
63           },
64           "average_score": 8.75
65        },
66        "final_score": 0.875
67    }
68
69    **Important Notes:**
70    - Please strictly follow the scoring criteria and avoid directly giving high scores (0.85+)
71    - You must complete attribute extraction and difference comparison before giving dimension scores
72    - If the modified diagram has obvious visual errors (such as element overlap, text misalignment), points
           should be deducted in the aesthetic quality dimension
73    - The final score must be calculated based on the average of dimension scores, not estimated directly
```

*Listing 8.* SCS Prompt for Task 2

### E.3.2. CODEVQA

**CodeVQA Evaluation Prompt**    **Purpose**: Answer CodeVQA questions based on `mxGraph` XML code.

```
1     You are a professional diagram analysis expert. Please carefully analyze the following Draw.io \texttt{mxGraph}
          XML code and answer ALL the questions below based on the \texttt{mxGraph} XML content.
2
3     **Generated \texttt{mxGraph} XML Code:**
4     ```xml
5     {generated_xml}
6     ```
7
8     **Questions:**
9     {questions_text}
10
11    **Instructions:**
12    - Analyze the \texttt{mxGraph} XML code structure, elements, attributes, and content
13    - For counting questions: Count specific elements or attributes in the \texttt{mxGraph} XML
14    - For identification questions: Find specific text values, IDs, or attributes in the \texttt{mxGraph} XML
15    - For relationship questions: Analyze connections, hierarchy, or relationships between elements in the \texttt
          {mxGraph} XML
16    - Answer each question based solely on the \texttt{mxGraph} XML code provided
17    - Do not make assumptions beyond what is in the \texttt{mxGraph} XML
18
19    **Output Format (JSON):**
20    Please provide answers in the following JSON format (one answer per question):
21    {
```

```
22        "Q1": "<answer to question 1>",
23        "Q2": "<answer to question 2>",
24        ...
25    }
26
27    **Important:**
28    - Answer each question directly without additional explanations
29    - Use the exact question numbers (Q1, Q2, Q3, etc.) as keys
30    - Provide concise answers only
31    - Base your answers strictly on the \texttt{mxGraph} XML code content
```

*Listing 9.* CodeVQA Evaluation Prompt

**CodeXQA Question-Answer Generation Prompt**   **Purpose**: Generate counting, identification, and relationship QA pairs for the CodeXQA benchmark based on the original image.

```
1     Please analyze the provided diagram image and generate 3 question-answer pairs for testing semantic
          information retention.
2     Must include the following three types, one of each:
3     1. Counting: Count the number of elements in the diagram
4     2. Identification: Identify attributes or labels of specific elements
5     3. Relationship: Identify relationships or connections between elements
6
7     Image: [Image Placeholder]
8
9     **Requirements (Enhanced Version):**
10
11    **1. Depth Requirements (Improve Discrimination)**
12    - **Counting questions**: Should not only ask "how many nodes" which is too simple. Should include counting of
          specific attributes, for example:
13      - "How many blue circular nodes are in the diagram?"
14      - "How many dashed connections are there?"
15      - "How many nodes have labels starting with the letter 'A'?"
16    - **Identification questions**: Should test specific visual details, for example:
17      - "What is the fill color of the node with id 'node_5'?"
18      - "What is the label of the node located at the top-left corner of the diagram?"
19      - "What color is the line connecting node A and node B?"
20    - **Relationship questions**: Should test multi-level connections or complex relationships, for example:
21      - "Through which intermediate nodes can node A reach node D?"
22      - "Which nodes are connected to both node B and node C?"
23      - "How many nodes are in the longest path from the root node to a leaf node?"
24
25    **2. Semantic Uniqueness**
26    - Ensure each question's answer is unique and unambiguous in the diagram
27    - Avoid questions that can be answered through common sense or context
28    - Questions should require the model to carefully observe the image to answer
29
30    **3. Visual Anchors**
31    - Guide questions to focus on key details and specific locations in the diagram
32    - Use specific identifiers (such as node IDs, position descriptions) to anchor questions
33    - Prevent models from answering through general reasoning rather than visual observation
34
35    **4. Verifiability**
36    - Answers should be specific and verifiable (numbers, color values, label text, etc.)
37    - Avoid subjective judgment questions
38
39    **Output Format (JSON):**
40    {
41      "qa_pairs": [
42        {
43          "question": "How many blue circular nodes are in the diagram?",
44          "answer": "3",
45          "question_type": "counting",
46          "visual_anchor": "blue circular nodes"
47        },
48        {
49          "question": "What is the fill color (in hexadecimal) of the node with id 'anchor'?",
50          "answer": "#FF5733",
51          "question_type": "identification",
52          "visual_anchor": "node ID 'anchor'"
53        },
54        {
55          "question": "Through which intermediate nodes can node A reach node D? Please list in order.",
56          "answer": "Node B and Node C",
57          "question_type": "relationship",
58          "visual_anchor": "path from node A to node D"
59        }
60      ]
```

```
61        }
```

*Listing 10.* Prompt for CodeXQA QA Generation

### E.3.3. XDRFR

**XDRFR Single Question Evaluation Prompt**   **Purpose**: Evaluate a single question based on the model's incremental modification output (JSON).

```
1     You are a professional \texttt{mxGraph} XML modification evaluation expert. Please answer the following
          question with "Yes" or "No" based on the provided model output JSON (containing incremental modifications)
          and modification instruction.
2
3     **Important Note**: The modification instruction uses natural language descriptions (based on rendered images).
          The model output JSON contains incremental modifications in the format of original_fragment and
          modified_fragment pairs. You need to understand the natural language instruction, then verify whether the
          modifications in the JSON satisfy the instruction requirements.
4
5     **Model Output JSON (Incremental Modifications):**
6     ```json
7     {model_output_json_str}
8     ```
9
10    **Modification Instruction (Natural Language):**
11    "{instruction}"
12
13    **Question:**
14    {question}
15
16    **Requirements:**
17    - Answer with only "Yes" or "No", do not output any other content
18    - Do not output reasons, explanations, evidence, or judgment logic
19    - Answer directly with Yes/No
20    - Must base judgment on the model output JSON content (incremental modifications), and verify whether the
          modifications satisfy the instruction requirements, not speculation
21
22    **Output Format (JSON):**
23    {
24        "answer": "Yes" or "No"
25    }
26
27    **JSON Format:**
28    Each modification object contains `original_fragment` (original \texttt{mxGraph} XML) and `modified_fragment`
          (modified \texttt{mxGraph} XML).
29
30    **Evaluation Process:**
31    1. Analyze each modification by comparing `original_fragment` and `modified_fragment`
32    2. Map instruction requirements to modifications
33    3. Verify completeness and correctness
34
35    **Verification Rules:**
36    - **Text changes**: Check `value` attribute in `modified_fragment`
37    - **Color changes**: Check `fillColor`/`strokeColor` in `style` attribute (hex codes like #0000FF or color
          names)
38    - **Position/Size**: Check `mxGeometry` coordinates/dimensions
39    - **Add/Remove**: Check for new elements in `modified_fragment` or empty `modified_fragment` for removals
40    - **Attributes**: Compare attributes between fragments
41
42    **Rules:**
43    - Analyze all modifications and verify they match the instruction
44    - Verify completeness (all instruction aspects addressed) and correctness (changes align with instruction)
45    - If uncertain, answer "No" (conservative strategy)
```

*Listing 11.* XDRFR Single Question Evaluation Prompt

**XDRFR Batch Evaluation Prompt**   **Purpose**: Batch evaluate multiple questions based on the model's output incremental modification JSON.

```
1     You are a professional XML modification evaluation expert. Please answer all the following questions with "Yes
          " or "No" based on the provided model output JSON (containing incremental modifications) and modification
          instruction.
2
3     **Important Note**: The modification instruction uses natural language descriptions (based on rendered images).
          The model output JSON contains incremental modifications in the format of original_fragment and
          modified_fragment pairs. You need to understand the natural language instruction, then verify whether the
          modifications in the JSON satisfy the instruction requirements.
```

```
4
5     **Model Output JSON (Incremental Modifications):**
6     ```json
7     {model_output_json_str}
8     ```
9
10    **Modification Instruction (Natural Language):**
11    "{instruction}"
12
13    **Question List:**
14    {questions_text}
15
16    **Requirements:**
17    - Answer each question with only "Yes" or "No", do not output any other content
18    - Do not output reasons, explanations, evidence, or judgment logic
19    - Answer each question directly with Yes/No
20    - Must base judgment on the model output JSON content (incremental modifications), and verify whether the
          modifications satisfy the instruction requirements, not speculation
21
22    **Output Format (JSON):**
23    {
24        "answers": [
25            {"question": "Complete text of question 1", "answer": "Yes"},
26            {"question": "Complete text of question 2", "answer": "No"},
27            ...
28        ]
29    }
30
31    **JSON Format:**
32    Each modification object contains `original_fragment` (original \texttt{mxGraph} XML) and `modified_fragment`
          (modified \texttt{mxGraph} XML).
33
34    **Evaluation Process:**
35    1. Analyze each modification by comparing `original_fragment` and `modified_fragment`
36    2. Map instruction requirements to modifications
37    3. Verify completeness and correctness
38
39    **Verification Rules:**
40    - **Text changes**: Check `value` attribute in `modified_fragment`
41    - **Color changes**: Check `fillColor`/`strokeColor` in `style` attribute (hex codes like #0000FF or color
          names)
42    - **Position/Size**: Check `mxGeometry` coordinates/dimensions
43    - **Add/Remove**: Check for new elements in `modified_fragment` or empty `modified_fragment` for removals
44    - **Attributes**: Compare attributes between fragments
45
46    **Rules:**
47    - Analyze all modifications and verify they match the instruction
48    - Verify completeness (all instruction aspects addressed) and correctness (changes align with instruction)
49    - If uncertain, answer "No" (conservative strategy)
```

*Listing 12.* XDRFR Batch Evaluation Prompt

### E.4. Instruction Templates

Expert-authored directives cover structural, stylistic, textual, and compositional edits:

- **Structural**: add/delete nodes or edges; re-parenting; grid/flow re-layout.

- **Stylistic**: palette changes; line/fill styles; font family/size/weight updates.

- **Textual**: rename labels; insert annotations; consolidate legends.

- **Compositional**: merge diagrams; extract subgraphs; duplicate and refactor patterns.

## F. Evaluation Methods and Metrics

Motivated by findings that LLMs degrade substantially when evaluated on *generating* coherent output rather than retrieving facts (Liu et al., 2024a), that single-metric evaluations miss multi-dimensional capability gaps (Dong et al., 2025), and that code generation is among the most degradation-prone fundamental abilities under resource-constrained deployment (Liu et al., 2026), we design a decomposed metric suite (CodeXQA, XDRFR, ESR, SCS) that jointly assesses structural accuracy, instruction compliance, and visual fidelity of generated mxGraph XML.

## F.1. Question Set Construction

### F.1.1. CODEXQA QUESTION TYPES

CodeXQA evaluates semantic retention through three distinct question types. Each sample in the benchmark contains exactly one question of each type.

- **Counting**: Evaluates the model's ability to aggregate statistics about specific visual elements.
  - *Example*: "How many blue circular nodes are in the diagram?"
  - *Focus*: Color, shape, and object detection.

- **Identification**: Tests the precise retrieval of attributes for specific elements.
  - *Example*: "What is the fill color (in hex) of the node with id 'node_5'?"
  - *Focus*: Fine-grained attribute recognition and text-to-object mapping.

- **Relationship**: Assesses the understanding of topological connections and flows.
  - *Example*: "Through which intermediate nodes can node A reach node D?"
  - *Focus*: Graph topology, pathfinding, and connectivity.

### F.1.2. XDRFR DECOMPOSITION LOGIC

For Task 2 (Instruction Editing), we employ a "Decompose-and-Verify" strategy. Natural language instructions are broken down into atomic, verifiable checkpoints.

**NL to `mxGraph` XML Mapping Rules:**

- **Text References**: "The 'Start' node" $\rightarrow$ `mxGraph` XML node with `value="Start"`.

- **Spatial References**: "The top-most node" $\rightarrow$ `mxGraph` XML node with $\min(y)$ in `mxGeometry`.

- **Style References**: "Red node" $\rightarrow$ `mxGraph` XML node with `fillColor="#FF0000"` in `style`.

- **Connection References**: "Link from A to B" $\rightarrow$ `mxCell` where `source` matches A's ID and `target` matches B's ID.

**Complexity-Adaptive Question Count:**

- **Simple Instructions** (1-2 atomic ops): Decomposed into 1-2 questions.

- **Medium Instructions** (3-4 atomic ops): Decomposed into 2-3 questions.

- **Complex Instructions** (5-7 atomic ops): Decomposed into 3-5 questions.

## F.2. Evaluation Metrics Details

We provide precise mathematical definitions for the key metrics used in VCG-Bench.

### F.2.1. TASK 1: GENERATION METRICS

**1. Execution Success Rate (ESR)** ESR measures the proportion of generated `mxGraph` XML codes that are syntactically valid and renderable.

$$\text{ESR} = \begin{cases} 1.0 & \text{if } \texttt{mxGraph XML\_valid} \wedge \text{Render\_success} \\ 0.0 & \text{otherwise} \end{cases}$$

**2. `mxGraph` XML Token Count (XTC)** XTC is a proxy for structural complexity, calculated using the 'cl100k_base' tokenizer. Primarily used for image difficulty classification, not as a model performance evaluation metric.

$$\text{XTC} = \text{Tokenize}(\texttt{mxGraph XML\_code}, \text{cl100k\_base}) \tag{4}$$

**3. Style Consistency Score (SCS)** SCS is a VLM-based (`gemini-3-pro-preview`) perceptual metric evaluating three dimensions on a 0-10 scale.

$$\text{SCS} = \frac{1}{30}\left(S_{\text{visual}} + S_{\text{layout}} + S_{\text{aesthetic}}\right) \tag{5}$$

Where:

- $S_{\text{visual}}$: Color palette, line styles, node shapes.

- $S_{\text{layout}}$: Spatial arrangement, alignment, spacing.

- $S_{\text{aesthetic}}$: Overall visual harmony and professional look.

**4. CodeXQA Accuracy** CodeXQA is the average accuracy across the three question types (Counting, Identification, Relationship).

$$\text{CodeXQA} = \frac{1}{N}\sum_{i=1}^{N}\mathbb{I}(\text{Match}(\text{Answer}_i^{\text{pred}}, \text{Answer}_i^{\text{gt}}))$$

Matching strategies include Exact Match, Inclusion, and Semantic Similarity.

**5. SigLIP2 Score (SigLIP2 Semantic Similarity Score)** This metric uses the SigLIP2 model to calculate semantic similarity between the original and generated images.

$$\text{SigLIP2} = \text{CosineSimilarity}(\text{SigLIP2}(\text{Original}), \text{SigLIP2}(\text{Generated}))$$

The model used is `google/siglip2-so400m-patch16-512`, with cosine similarity in the range 0–1.

F.2.2. TASK 2: EDITING METRICS

**1. XDRFR (XML Decomposed Requirements Following Ratio)** XDRFR is the primary metric for instruction following; it calculates the pass rate of decomposed Yes/No questions.

$$\text{XDRFR} = \frac{1}{M}\sum_{i=1}^{M}\mathbb{I}(\text{Answer}_i = \texttt{"Yes"})$$

Where $M$ is the number of decomposed questions for the instruction. The evaluation is performed purely on the XML text, avoiding visual rendering artifacts.

**2. SCS for Task 2** This metric is adapted to focus on style preservation during edits.

$$\text{SCS}_{\text{Task2}} = \frac{1}{20}\left(S_{\text{style}} + S_{\text{aesthetic}}\right)$$

Crucially, this metric does not penalize content changes (which are intended) but ensures the *style* remains consistent with the original diagram.

**3. `mxGraph` XML Edit Distance (XED)** XED calculates the edit distance between the original `mxGraph` XML and the modified `mxGraph` XML, quantifying the magnitude of code-level modifications.

$$\text{XED} = \text{LevenshteinDistance}(\text{Original } \texttt{mxGraph} \text{ XML}, \text{Modified } \texttt{mxGraph} \text{ XML})$$

It uses the standard Levenshtein distance algorithm, based on character-level comparison of `mxGraph` XML strings.

F.2.3. EVALUATION METRICS FORMALIZATION

We compute layout-aligned measures between reference ($R$) and candidate ($C$) outputs.

- Style Consistency: attribute agreement rate across evaluated fields,

$$\text{Style} = \frac{\sum_i \mathbf{1}\big[\text{attr}_i^{(r)} = \text{attr}_i^{(c)}\big]}{\# \text{ attributes}}.$$

- Editability: proportion of outputs that (i) load without errors and (ii) support move/resize and re-routing without breaking attachments.

- Instruction Compliance: ratio of directives satisfied, combining rule-based checks with adjudication on ambiguous cases.

A weighted aggregate prioritizes structural fidelity and completeness while incorporating style, editability, and compliance.

### F.3. Validation and Reproducibility

We use a validator pipeline to ensure schema validity and deterministic metric computation.

**VCG-Bench Validator Pipeline:**

**Algorithm: VCG-Bench Validator Pipeline**

1. **Input**: candidate `mxGraph` XML $C$, reference `mxGraph` XML $R$, directives $D$

2. Parse and validate schema($C$); report well-formedness and loadability

3. Render $C$ and $R$ under identical settings for spatial comparison

4. Detect elements and align via category + proximity; perform bipartite matching

5. Compute IoU, completeness, style consistency

6. Run editability checks: move/resize nodes, re-route connectors

7. Evaluate directive compliance from $D$ (presence/absence, counts, layout constraints)

8. **Output**: per-metric scores and weighted aggregate

