# OpenReview forum: "VCG-Bench: Towards A Unified Visual-Centric Benchmark for Structured Generation and Editing"
_ICML.cc/2026/Conference — ICML 2026 regular_

### Official Review · Reviewer_n55j · 2026-03-07

**Soundness:** 3
**Presentation:** 3
**Significance:** 3
**Originality:** 3
**Overall Recommendation:** 4
**Confidence:** 4

**Summary:**

The paper introduces VCG-Bench, a benchmark for evaluating vision-language models on diagram generation and editing by representing diagrams as structured mxGraph XML code. It provides a dataset of diagram images and editing instructions, with evaluation metrics to assess models’ ability to generate and modify structured diagram code from visual inputs.

**Compliance With Llm Reviewing Policy:**

Affirmed.

**Final Justification:**

The authors have addressed my concerns in the rebuttal period, and I will raise my score.
I also encourage the authors to ensure that all revisions and clarifications are clearly reflected in the final version of the paper.

**Key Questions For Authors:**

Please see weakness.

**Limitations:**

Please see weakness.

**Strengths And Weaknesses:**

**strength**:

The paper presents a well-designed benchmark construction pipeline with multiple quality control stages. The paper is easy to read and well organized. The figures are clear and visually appealing.

The topic is very interesting. Evaluating vision-language models on structured diagram generation and editing is relatively underexplored compared to other multimodal tasks.

The experimental evaluation is extensive and compares a wide range of both closed-source and open-source models. The results reveal several interesting findings, such as the difficulty models have in correctly parsing structural elements (especially counting and instance grounding).

**weakness**:

The paper claims to introduce a “visual-centric” benchmark. However, Task 2 (Code-to-Code editing) appears to rely primarily on XML manipulation and code-level verification rather than visual reasoning. Could the authors clarify how the visual is essential in this task?

about eval: Both Task 1 and Task 2 produce mxGraph XML outputs. However, several evaluation metrics (e.g., SCS and SigLIP2 similarity) require rendered diagrams. The paper does not explicitly describe the rendering pipeline (e.g., which renderer or configuration is used). Could the authors clarify how XML is rendered during evaluation?

about eval metrics: The paper emphasizes topology and structural correctness, but the evaluation metrics rely primarily on embedding similarity and QA-style checks rather than explicit graph topology metrics. Could the authors clarify how exact structural correctness is measured?

Some details are missing in the paper:
- The paper mentions human verification in the data curation pipeline, but details about the annotators are missing (e.g., number of annotators, expertise, and annotation guidelines).
- The pipeline relies on Gemini-3-Pro to generate structured JSON captions. How do the authors ensure the correctness of these captions? Are they manually verified or evaluated quantitatively?
- In section 3.2, the paper states that “a VLM generates mxGraph XML from images and captions,” but the specific model is not described. According to appendix B.3, is Gemini-3-Pro still used in this process?
- All model-based evaluations employ Gemini-3-Pro for consistency. It would be helpful to report the correlation between Gemini-based evaluation and human judgments.

some relevant papers about diagram/SVG generation and editing benchmark are missing:

[1] Wei J, Tan C, Chen Q, et al. From words to structured visuals: A benchmark and framework for text-to-diagram generation and editing[C]//Proceedings of the Computer Vision and Pattern Recognition Conference. 2025: 13315-13325.

[2] Zhuo L, Han S, Pu Y, et al. Factuality Matters: When Image Generation and Editing Meet Structured Visuals[J]. arXiv preprint arXiv:2510.05091, 2025.

[3] Cui Z, Yuan J, Wang H, et al. Draw with Thought: Unleashing Multimodal Reasoning for Scientific Diagram Generation[C]//Proceedings of the 33rd ACM International Conference on Multimedia. 2025: 5050-5059.

[4] Nishina K, Matsui Y. SVGEditBench: A Benchmark Dataset for Quantitative Assessment of LLM's SVG Editing Capabilities[C]//Proceedings of the IEEE/CVF Conference on Computer Vision and Pattern Recognition. 2024: 8142-8147.


about presentation:
- In line 83 of the intro, the abbreviation Vision-Language Models (VLMs) is introduced again even though it has already been defined earlier. The full name may not be necessary here.
- citation of Claude is missing in line 083 page 2
- SALT-NLP/Design2Code dataset mentioned on page 4 lacks a proper citation.
- The second-to-last paragraph of the introduction feels somewhat abrupt. The paper starts discussing the generation pipeline before clearly introducing what VCG-Bench is. It might improve readability if the description of the pipeline is moved to another proper place.

---

> ### Author Rebuttal · Authors · 2026-03-30
>
> **To Reviewer n55j**
>
> Thank you for the detailed comments on evaluation, data construction, and related work. We briefly respond below.
>
> **Q1: Whether Task 2 is “visual-centric”**
> **A1:** Although Task 2 is implemented as XML patch editing, the edited object is still the structured representation of the same diagram. The rendered image view and the mxGraph XML are tightly bound through Draw.io's deterministic rendering pipeline. The edited XML is then re-rendered and evaluated with SCS, so the final judgment falls on the visual result rather than on code manipulation detached from visual semantics.
>
> **Q2: XML rendering pipeline and renderer**
> **A2:** We uniformly use the official diagrams.net/draw.io CLI export (-x mode, mxGraph+Chromium) to keep rendering consistent with what annotators see. We also compared other headless exports and native backends, but they may introduce small differences relative to the official editor. Therefore, all main experiments use the official CLI with a unified export configuration.
>
> **Q3: How exact structural correctness is measured**
> **A3:** For complex layouts, explicit graph-theoretic metrics alone cannot cover all variants. In structured-visual evaluation, it is common to combine QA/probing (e.g., VCode[1], MMMU[2], MMMU-Pro[3]) with embedding-based similarity. Our CodeXQA and XDRFR directly evaluate code and incremental semantics, while SigLIP2/SCS complements them from visual layout and appearance. Appendix Table 9 reports a manual audit of XDRFR, showing that the automatic evaluation is relatively reliable.
>
> **Q4: Human verification and annotator details**
> **A4:** Data curation and human verification were conducted by 6 project members (4 PhD students, 2 Master's students in AI). We prepared an internal written guideline: first exclude screenshots, photos, pure statistical charts, and samples with unclear main structure; during verification, check structural correctness first (node/edge missing, wrong links, topological errors), then text and style; Task 2 instructions must use natural-language references visible in the rendered image, avoid ids/coordinates/XML attributes, and be decomposable into clear atomic operations. Suspicious cases are cross-reviewed and removed when necessary.
>
> **Q5: How the quality of structured JSON captions is ensured**
> **A5:** We do not directly accept Gemini-generated captions. We first perform a quality screening on the VLM-restored image; when suspicious cases (missing elements, messy layouts) are found, annotators re-check the image against the JSON description and revise when necessary. In short, automatic screening + human review + manual correction.
>
> **Q6: Which model is used for “VLM generates XML” in Section 3.2**
> **A6:** We will distinguish data construction from formal evaluation in the revision. In the current pipeline, the structured caption is fixed to Gemini; candidate XML synthesis was tried with multiple VLMs, after which high-quality samples were retained through executability checks, rendering consistency checks, and human screening. In the formal benchmark evaluation, we compare multiple models on Task 1/Task 2 rather than evaluating Gemini alone.
>
> **Q7: Correlation between Gemini-based evaluation and human judgment**
> **A7:** Fig.6 shows the correlation between SCS and human rankings, and Appendix Table 9 reports a manual audit of XDRFR. Together, these results indicate that although Gemini is not perfect as a unified judge, its scores are reasonably consistent with human judgment.
>
> **Q8: Missing relevant papers on diagram/SVG generation and editing benchmarks**
> **A8:** Thank you for the suggestions. In the revision, we will add the works you pointed out in Related Work, including From Words to Structured Visuals, Factuality Matters, Draw with Thought, SVGEditBench[4-7], and SVGEditBench V2[8].
>
> **Q9: Presentation and structure issues**
> **A9:** We will streamline wording (e.g., remove repeated full expansion of VLMs in the introduction), add missing citations (Claude, Design2Code), and restructure the introduction to introduce VCG-Bench before describing the data construction pipeline.
>
> **References (titles only):**
> [1] VCode: a Multimodal Coding Benchmark with SVG as Symbolic Visual Representation
> [2] Mmmu: A massive multi-discipline multimodal understanding and reasoning benchmark for expert agi
> [3] Mmmu-pro: A more robust multi-discipline multimodal understanding benchmark
> [4] From Words to Structured Visuals: A Benchmark and Framework for Text-to-Diagram Generation and Editing
> [5] Factuality Matters: When Image Generation and Editing Meet Structured Visuals
> [6] Draw with Thought: Unleashing Multimodal Reasoning for Scientific Diagram Generation
> [7] SVGEditBench: A Benchmark Dataset for Quantitative Assessment of LLM's SVG Editing Capabilities
> [8] SVGEditBench V2: A Benchmark for Instruction-based SVG Editing

---

> > ### Author Rebuttal · Reviewer_n55j · 2026-04-03
> >
> > My previous concerns have been satisfactorily resolved.

---

> > > ### Author Response · Authors · 2026-04-03
> > >
> > > Dear Reviewer n55j,
> > >
> > > Thank you for your positive acknowledgment of our rebuttal and for confirming that your previous concerns have been fully resolved.
> > >
> > > Given that your concerns have been fully resolved, may we kindly ask whether you would consider updating your overall recommendation accordingly?
> > >
> > > Thank you again for your valuable time and constructive feedback.
> > >
> > > Sincerely,
> > > The Authors

---

### Official Review · Reviewer_S7dz · 2026-03-11

**Soundness:** 2
**Presentation:** 3
**Significance:** 3
**Originality:** 2
**Overall Recommendation:** 3
**Confidence:** 3

**Summary:**

This paper presents VCG-Bench, a unified visual-centric benchmark for structured diagram generation and editing. It adopts mxGraph XML as a “Diagram-as-Code” representation and jointly evaluates both Vision-to-Code diagram generation and Code-to-Code instruction-based editing. The authors construct a dataset of 1,449 diagrams spanning 6 domains and 15 sub-domains, and introduce multi-dimensional evaluation metrics including ESR, SCS, CodeXQA, and XDRFR. Experiments are conducted on a range of both closed-source and open-source models.

**Compliance With Llm Reviewing Policy:**

Affirmed.

**Final Justification:**

The rebuttal provides additional clarifications on benchmark construction, evaluation settings, and comparisons, which help improve the overall clarity of the work. However, my main concerns remain: the novelty is still limited as the contribution is largely centered on dataset and benchmark integration, and the advantages over existing benchmarks are not yet fully convincing. While the added explanations are helpful, they do not significantly change my overall assessment, so I maintain my original score recommendation.

**Key Questions For Authors:**

N/A

**Limitations:**

yes

**Strengths And Weaknesses:**

Strengths
1. The paper focuses on structured diagram generation and editing in professional workflows, which is distinct from standard natural-image or general-purpose VLM benchmarks and has clear practical value.
2. The dataset has reasonably broad coverage. The 1,449 samples span multiple domains such as academic, software, business, and UI/UX, and are further stratified by difficulty.
3. The paper provides relatively rich implementation details, which is helpful for reproducibility and for the community to build upon this benchmark.

Weaknesses
1. The core novelty is incremental and leans more toward benchmark engineering and integration; the contribution to learning frameworks, model architectures, or stronger automated construction methods is limited.
2. The discussion of related work remains somewhat weak, especially regarding graph-structured / diagram visual reasoning benchmarks, symbolic visual representation benchmarks, and top-down or abstract visual reasoning benchmarks.
3. The dataset scale is still limited. While 1,449 diagrams and 1,005 editing instructions are sufficient for an initial evaluation, they may be insufficient to support the long-term impact of a “unified benchmark.” The scale and diversity remain limited, particularly in terms of non-English content, complex professional symbols, and highly rare layout patterns.

---

> ### Author Rebuttal · Authors · 2026-03-30
>
> **To Reviewer S7dz**
>
> Thank you for recognizing the value of our work in professional diagram workflows, cross-domain data, and implementation details. We briefly respond to the three concerns below.
>
> **Q1: Novelty positioning and “automatic construction”**
> **A1:** We agree the contribution is not a new learning framework or model backbone, but a reproducible, executable, and editable benchmark built around the real Draw.io/mxGraph workflow. However, this work is more than a simple integration of existing pieces:
> - **Target format focus**: To our knowledge, existing benchmarks have not systematically used mxGraph XML as the target format while jointly evaluating vision-to-code generation and structured editing. This work makes the first systematic construction along this combined direction.
> - **Data construction**: We provide a Task 1 pipeline for synthesizing and filtering high-quality mxGraph XML: localize diagram regions from PDF/PNG/PPT → generate structured JSON descriptions → synthesize mxGraph XML → filter with ESR, SCS, CodeXQA, and manual checking.
> - **Task positioning**: Task 1 and Task 2 are not just two stacked subtasks, but correspond to the two core capabilities of a Draw.io agent: Task 1 converts diagrams in images/PDFs/PPTs into editable mxGraph XML (input and state construction); Task 2 performs precise incremental edits over existing structure based on user instructions (execution). Together, they form the basic loop of “understand the diagram → convert to code → continue editing,” providing a clear capability decomposition, data foundation, and evaluation interface for future diagram/Draw.io agents.
>
> **Q2: Insufficient coverage of related work**
> **A2:** We agree the related work can be further strengthened. However, the current version does have some coverage:
> - chart/graph visual reasoning (ChartQA, CharXiv[1-2]);
> - symbolic visual representation (VCode, SVGenius, DeepSVG[3-5]);
> - abstract multimodal evaluation (MMMU, MMMU-Pro, MM-Vet[6-8]).
>
> So the issue is more about organization and elaboration than absence. Following your suggestion, we will add several more directly relevant works in the revision, including From Words to Structured Visuals, Factuality Matters, Draw with Thought, SVGEditBench[9-12], and reorganize related work along three axes—chart/graph reasoning, symbolic visual representation, and abstract multimodal reasoning—to better clarify our relation to adjacent directions.
>
> **Q3: Scale, collection cost, and diversity**
> **A3:** We agree the benchmark can be further expanded, but constructing high-quality data of this kind is expensive.
> - **Cost**: We agree that scale remains a limitation. In this first version, we prioritized quality and executability, which also reflects the high construction and verification cost (Task 1 synthesis costs about 20,457 tokens per sample on average, Task 2 editing costs about 11,892 tokens per instruction, and each sample requires multiple model runs and manual screening). We plan to further expand the dataset in our next agent work.
> - **Diversity**: Regarding non-English content, complex professional symbols, and rare layout patterns, the current dataset is not entirely missing them. The current sample pool is roughly 7:3 English vs. Chinese; some images come directly from papers and technical materials, so they include formulas, professional symbols, and more complex visual elements, which Draw.io/mxGraph can also support reasonably well. Meanwhile, the dataset covers 15 sub-domains and multiple layout types such as posters, PPT slides, flowcharts, and scientific diagrams (see Section 3.3 and Fig.4).
> - **Future expansion**: Of course, extremely long-tail and very rare layouts are still underrepresented in this first version, and that remains an important direction for future expansion.
>
> **References (titles only):**
> [1] ChartQA: A Benchmark for Question Answering about Charts with Visual and Logical Reasoning
> [2] CharXiv: Charting Gaps in Realistic Chart Understanding in Multimodal LLMs
> [3] VCode: a Multimodal Coding Benchmark with SVG as Symbolic Visual Representation
> [4] SVGenius: Benchmarking LLMs in SVG Understanding, Editing and Generation
> [5] DeepSVG: A Hierarchical Generative Network for Vector Graphics Animation
> [6] Mmmu: A massive multi-discipline multimodal understanding and reasoning benchmark for expert agi
> [7] Mmmu-pro: A more robust multi-discipline multimodal understanding benchmark
> [8] MM-Vet: Evaluating Large Multimodal Models for Integrated Capabilities
> [9] From Words to Structured Visuals: A Benchmark and Framework for Text-to-Diagram Generation and Editing
> [10] Factuality Matters: When Image Generation and Editing Meet Structured Visuals
> [11] Draw with Thought: Unleashing Multimodal Reasoning for Scientific Diagram Generation
> [12] SVGEditBench: A Benchmark Dataset for Quantitative Assessment of LLM's SVG Editing Capabilities

---

> > ### Author Rebuttal · Reviewer_S7dz · 2026-04-03
> >
> > Thank you to the authors for the rebuttal.
> >
> > My concerns are only partially resolved:
> > (1) The response clarifies the benchmark positioning, but it does not change my view that the main contribution remains largely benchmark engineering and integration rather than a stronger methodological advance;
> > (2) The related work discussion is improved, but this is mainly a matter of adding and reorganizing references, and the gap in positioning relative to adjacent structured visual reasoning benchmarks is only partially addressed;
> > (3) The explanation for dataset scale and diversity is helpful, but the benchmark still appears limited in size for a “unified benchmark,” and long-tail coverage remains insufficient, especially for rarer layouts and broader symbol diversity.
> >
> > Overall, I find the concerns partially addressed but not fully resolved, and I will maintain my original score.

---

> > > ### Author Response · Authors · 2026-04-05
> > >
> > > **To Reviewer S7dz:**
> > >
> > > Thank you for your follow-up response and for your valuable time. We deeply respect your perspective and appreciate the high standards you hold for this work. Given a potential misalignment between expectations for this specific domain and the current landscape of multimodal benchmarks, we would like to provide some objective clarifications regarding the dataset scale and our original intentions.
> > >
> > > **1. On Dataset Scale and Diversity:**
> > > We fully agree that "larger is always better." However, in the domain of fine-grained, structured visual-code generation, 1,449 highly-verified, complex mxGraph XML samples actually represents a substantial push forward in scale.
> > >
> > > Unlike programmatically generated charts (e.g., Matplotlib charts generated via simple scripts, which can easily reach thousands of samples; see ChartMimic[1]), Draw.io diagrams require capturing complex, non-rigid topologies. The data generation and verification costs are exceptionally high (our samples average 20,457 tokens). Compared to recent benchmarks in similar complex structured domains, our scale is highly competitive:
> > > * **Plot2Code[2] (2025):** 368 samples
> > > * **VCode[3] (SVG, 2025):** 464 samples
> > > * **Design2Code[4] (UI, 2025):** 484 samples
> > > * **PlotCraft[5] (2025):** 982 samples
> > > * **MM-Vet[6] (2023):** 218 samples
> > > * **VCG-Bench (Ours):** 1,449 samples (spanning 15 sub-domains)
> > >
> > > Given the high token cost and the strict manual verification required to ensure executability, these 1,449 samples provide a very solid starting point for this modality. Furthermore, as you kindly acknowledged in your initial "Strengths" section, our dataset already achieves "reasonably broad coverage... spanning multiple domains."
> > >
> > > **2. On Benchmark Engineering and Research Contribution:**
> > > We completely understand and appreciate your expectation for a "stronger methodological advance." When undertaking this work, our core original intention was to fill the evaluation gap for a highly challenging modality that previously lacked a benchmark: "Diagram-as-Code" via mxGraph. We sincerely hope that by systematically designing this unified, reproducible, and executable evaluation paradigm, we can provide a solid data infrastructure for the community. As demonstrated by our experiments, clearly exposing the precise bottlenecks of frontier models in topological reasoning contributes a modest but essential foundational step toward future substantive improvements in model architectures.
> > >
> > > Thank you again for your time and professional feedback. We will ensure these comparative dataset statistics are explicitly detailed in the final manuscript to better contextualize the scale and positioning of our work for future readers.
> > >
> > > **References**
> > >
> > > [1] Chartmimic: Evaluating lmm's cross-modal reasoning capability via chart-to-code generation
> > > [2] Plot2code: A comprehensive benchmark for evaluating multi-modal large language models in code generation from scientific plots
> > > [3] VCode: a Multimodal Coding Benchmark with SVG as Symbolic Visual Representation
> > > [4] Design2code: Benchmarking multimodal code generation for automated front-end engineering
> > > [5] PlotCraft: Pushing the Limits of LLMs for Complex and Interactive Data Visualization
> > > [6] Mm-vet: Evaluating large multimodal models for integrated capabilities

---

### Official Review · Reviewer_GbwP · 2026-03-12

**Soundness:** 2
**Presentation:** 3
**Significance:** 2
**Originality:** 3
**Overall Recommendation:** 4
**Confidence:** 4

**Summary:**

This paper introduces VCG-Bench, a benchmark for structured diagram generation and editing in mxGraph/Draw.io format. The core idea is to move away from pixel-space outputs and instead evaluate models on editable symbolic code. The benchmark has two tasks: (1) vision-to-code generation, where a model converts a diagram image into mxGraph XML, and (2) instruction-based editing, where a model modifies existing mxGraph code through localized patches. The dataset contains 1,449 diagrams across 6 domains and 15 sub-domains, and the paper proposes an evaluation suite built around executability, visual/style consistency, semantic QA over code, and instruction-following in code editing. The main empirical finding is that current frontier models do much better on code-to-code editing than on vision-to-code generation, and that open models struggle especially on producing executable XML.

**Compliance With Llm Reviewing Policy:**

Affirmed.

**Final Justification:**

I have considered the authors’ rebuttal carefully. The response addresses the main questions I had and provides helpful clarification on several technical points. While some minor limitations remain, they do not substantially affect my overall assessment. I therefore keep my original recommendation unchanged.

**Key Questions For Authors:**

1. How much does Task 1 performance change if you remove the Gemini-generated structured descriptions and evaluate on a random raw image input? If they change a lot, then the current task should be reframed more narrowly.

2. Why is Gemini used as a shared assistant/evaluator across the pipeline, and what safeguards did you use to reduce model-family bias?

3. Do you have harder Task 2 settings that are more discriminative, such as multi-step edits with ambiguous references, overlapping constraints, or edits that require graph-level reorganization? Right now Task 2 looks useful but a bit saturated. Evidence that the benchmark can separate models more strongly would improve the paper’s significance.

4. Can you compare against at least one non-LLM or pipeline baseline for Task 1, such as OCR/layout extraction plus rule-based graph recovery? This would help place the results in context and show whether the benchmark is testing genuinely hard multimodal reasoning rather than just XML formatting skill.

**Limitations:**

Yes.

**Strengths And Weaknesses:**

1. Soundness:
The paper is built around a reasonable benchmark design, and the overall setup is coherent. The task formulation is practical, the split between generation and editing is well motivated, and the evaluation includes criteria that matter for structured outputs, especially executability. My main concern is that the benchmark pipeline depends heavily on the same model family for data processing, task assistance, and evaluation. This makes it harder to separate true task difficulty from artifacts of the evaluation setup.
In particular, if Task 1 relies on a rendered model-generated structured descriptions of the raw image, then the setting is no longer a clean test of direct vision-to-code generation. That does not make the task invalid, but it does narrow what the reported results actually show. I also would have liked to see stronger validation of the automatic metrics, especially since they carry much of the empirical argument.

2. Presentation:
The paper is generally well written and easy to follow. The motivation is clear, the benchmark setup is explained in a sensible order, and the distinction between the two tasks helps the narrative. The figures and tables also do useful work.

3. Significance:
The paper tackles a real gap. Most multimodal benchmarks still treat diagrams as images, even when the practical use case is editable structure. Framing the problem around mxGraph is sensible for workflows where users actually want diagrams they can revise, not just screenshots. That makes the benchmark more practically grounded than many image-only settings. However,modeling the semantic hierarchy and explicit topology of diagrams is not a new challenge in symbolic graphics generation. I also do not see a large conceptual difference between handling SVG and mxGraph representations, since both are hierarchical XML-based formats with structured primitives and relations. So the main contribution seems to lie less in introducing a new underlying task and more in defining a benchmark around a specific editable diagram representation. Even with the current weaknesses, I can still see the benchmark being useful to the community, especially if the release is complete and the evaluation is strengthened.

4. Originality:
The paper has a clear point of novelty in how it packages the problem. A benchmark centered on editable mxGraph XML, with both diagram generation and instruction-based editing, is a useful and fairly distinctive contribution. The emphasis on executable, revisable diagram structure also gives the work a practical angle that helps it stand out from image-only benchmarks. At the same time, I do not think the underlying conceptual problem is entirely new. Representing visual artifacts through structured code, hierarchy, and relations has already been explored in nearby areas such as symbolic graphics generation, SVG generation, chart-to-code, and UI-to-code. For that reason, I do not see the paper as introducing a fundamentally new task class. The novelty is more specific: it comes from the benchmark design, the choice of representation, and the focus on editable diagrams as a target format.

---

> ### Author Rebuttal · Authors · 2026-03-30
>
> **To Reviewer GbwP**
>
> Thank you for recognizing the motivation, task design, and practical value of our benchmark. We briefly respond below.
>
> **Q1: Performance change after removing structured descriptions and task positioning**
> **A1:** Prior work such as Section 3.2 of ChartMimic[1] and Section 3 of VAGEN[2] suggests that structured descriptions help model understanding, so our main setting uses image+structured caption. We added an ablation on a random n=100 subset:
>
> | Setting | ESR | SCS | CODEXQA |
> | --- | --- | --- | --- |
> | With caption (original setting) | 1.000 | 0.805 | 0.930 |
> | Without caption | 0.980 | 0.690 | 0.786 |
>
> The drops in ESR, SCS, and CODEXQA confirm caption helps vision-to-mxGraph. We agree the task description should be more precise: main tables correspond to image+structured caption→mxGraph; raw image is a harder setting. We will clarify this in revision and add the ablation to appendix.
>
> **Q2: Why Gemini is used extensively and how to mitigate model-family bias**
> **A2:** We ran small pilots with multiple models and combined them with manual checks. Overall, Gemini performed relatively better in the pilots and manual spot checks, so we selected it as the current data-assistance model. To mitigate model-family bias, we employ manual verification and cross-checking of suspicious samples throughout the pipeline. This practice is also adopted in prior work, e.g., Self-Instruct[3], UltraFeedback[4], and Chart2Code53[5], where strong closed-source models are used as assistants alongside human validation.
>
> **Q3: Harder and more discriminative Task 2 settings (multi-step, ambiguity, diagram-level reorganization)**
> **A3:** We agree that a harder and more discriminative Task 2 setting would further increase the benchmark's importance. At the same time, we note that the current Task 2 still has limited headroom in some settings, but constrained incremental editing itself remains a common and core interaction mode for AI coding assistants (as reflected in code-editing benchmarks such as EDIT-Bench, CodeEditorBench[6-7]). The relatively high Task 2 scores also indicate that models are already relatively mature at incremental editing over existing structure, providing a foundation for future Draw.io agents.
>
> - Multi-step editing: we already include it in Section 4.3 through Easy/Medium/Hard settings defined by atomic operation count.
> - Ambiguous references: we agree they are important, but are better modeled as mixed-initiative clarification/planning rather than one-shot editing; this is beyond the scope of the current static benchmark, and Teaching Vision-Language Models to Ask[8] points in the same direction.
> - Diagram-level reorganization: we did consider instructions such as compact layout or lighter colors, but objective evaluation is difficult, so the main set prioritizes objectively verifiable rule-based edits. As a supplement, we provide 7 compact-layout examples at <https://anonymous.4open.science/r/task2_compact_layout-7DCD/>.
>
> **Q4: Non-LLM / pipeline baseline for Task 1**
> **A4:** We added Paper2Any[9] as a representative pipeline baseline (OCR/detection/layout analysis/rule recovery), not an end-to-end VLM. Its weakness is coarse decomposition: often treating the whole figure as a single block or performing coarse rectangular partitioning, outputting low-granularity patches in PPT rather than editable structured objects in Draw.io. Anonymous repo: https://anonymous.4open.science/r/task1_paper2any-E33F/. This shows Task 1 tests internal structure and editable semantics recovery, not just XML formatting.
>
> **References (titles only):**
> [1] ChartMimic: Evaluating LMM's Cross-Modal Reasoning Capability via Chart-to-Code Generation
> [2] VAGEN: Reinforcing World Model Reasoning for Multi-Turn VLM Agents
> [3] Self-Instruct: Aligning Language Models with Self-Generated Instructions
> [4] UltraFeedback: Boosting Language Models with Scaled AI Feedback
> [5] Chart2Code53: A Large-Scale Diverse and Complex Dataset for Enhancing Chart-to-Code Generation
> [6] EDIT-Bench: Evaluating LLM Abilities to Perform Real-World Instructed Code Edits
> [7] CodeEditorBench: Evaluating Code Editing Capability of Large Language Models
> [8] Teaching Vision-Language Models to Ask: Resolving Ambiguity in Visual Questions
> [9] Paper2Any: https://github.com/OpenDCAI/Paper2Any

---

> > ### Author Rebuttal · Reviewer_GbwP · 2026-04-03
> >
> > Thanks for the rebuttal. It improves the paper, but only partially addresses my concerns.
> >
> > The caption ablation is useful and should be added, but it also confirms that Task 1 is substantially easier with the structured caption. The Paper2Any baseline is a helpful addition and strengthens the empirical context. It supports the claim that the task is about recovering editable structure, not just formatting XML.
> >
> > My main concern about heavy Gemini dependence across the pipeline is still not fully resolved. Manual checks and precedent are not enough without stronger quantitative validation of bias and metric reliability. On Task 2, I understand the scope choice, but the rebuttal mostly reinforces my view that the current editing setting is useful yet somewhat saturated and not maximally discriminative.
> >
> > Overall, the rebuttal strengthens the submission, but not enough to materially change my assessment. I remain at weak accept.

---

> > > ### Author Response · Authors · 2026-04-07
> > >
> > > **To Reviewer GbwP:**
> > >
> > > Thank you for your follow-up response. We are glad that you found the caption ablation and the Paper2Any baseline helpful; we will incorporate both into the revision. Below we further clarify the two concerns that still remain.
> > >
> > > **1. On Gemini family bias and evaluation reliability.**
> > > We agree that this requires stronger quantitative validation. At the same time, we would like to clarify that not all metrics are equally exposed to potential judge-family bias. For Task 1, **ESR** is an executability check, while **CodeXQA** and **XDRFR** are based on objective question answering / incremental verification over structured code, and are therefore closer to standard-answer accuracy evaluation. The metric most likely to be affected by judge choice is **SCS**, which is a judge-based visual score. To address this, we conducted an additional **4×3 cross-judge validation** on **100 randomly sampled Task-1 examples**: we fixed the same set of Task-1 outputs and evaluated **4 generator models** (gpt-5.2, gemini-3-pro-preview, claude-sonnet-4-5, and GLM-4.6V) with **3 different judge models** (Gemini / GPT / Claude). Since we only kept cases where all three judges returned valid and directly comparable scores, the effective sample size for the table below is **n=78**.
> > >
> > > | Generator Model | Gemini Judge | GPT Judge | Claude Judge |
> > > | --- | --- | --- | --- |
> > > | gpt-5.2 | 0.626 ± 0.208 | 0.688 ± 0.121 | 0.635 ± 0.172 |
> > > | gemini-3-pro-preview | 0.788 ± 0.148 | 0.785 ± 0.095 | 0.754 ± 0.105 |
> > > | claude-sonnet-4-5 | 0.705 ± 0.186 | 0.748 ± 0.118 | 0.690 ± 0.144 |
> > > | GLM-4.6V | 0.415 ± 0.184 | 0.572 ± 0.153 | 0.465 ± 0.182 |
> > >
> > > More importantly, the **ranking order is identical across all three judges**: all three columns yield the same ordering, namely **gemini-3-pro-preview > claude-sonnet-4-5 > gpt-5.2 > GLM-4.6V**. In other words, while the **absolute scores** show some calibration differences across judges, the **relative ranking remains unchanged**. This is also consistent with the cross-judge agreement statistics: the Spearman correlation is **0.770** for Gemini vs GPT, **0.735** for Gemini vs Claude, and **0.721** for GPT vs Claude, with an average of **ρ = 0.742** (all highly significant, `p < 1e-50`). We also added **GLM-4.6V**, which does not belong to the OpenAI / Google / Anthropic families, as an external reference. If there were strong family bias, its ranking should fluctuate more across judges; instead, all three judges consistently rank it last. Therefore, while different judges may assign somewhat different **absolute values**, our main conclusions do not depend on Gemini alone. In the revision, we will include this cross-judge result and more clearly distinguish between “objective structural metrics” and “judge-based visual metrics.”
> > >
> > > **2. On Task 2 and its current discriminative power.**
> > > Thank you for your understanding of our scope choice and the practical value of the current editing setup. We would only like to offer one brief clarification here: the Task-2 results mainly suggest that current frontier models have become relatively mature at **constrained incremental editing given a complete structure**, and this is one of the capability boundaries that we intended Task 2 to capture. We agree that the harder and more discriminative settings you mentioned are valuable future directions, but they are beyond what the current version can cover and evaluate objectively in a reliable way. We will therefore calibrate the wording in the revision and define the scope of Task 2 more carefully.
> > >
> > > Overall, we appreciate your high standards regarding evaluation robustness and Task-2 discriminativeness. In the revision, we will add the cross-judge quantitative validation above and calibrate the Task-2 claims accordingly, so that the paper is more tightly aligned with the current evidence.

---

### Official Review · Reviewer_TQjD · 2026-03-12

**Soundness:** 4
**Presentation:** 4
**Significance:** 4
**Originality:** 3
**Overall Recommendation:** 4
**Confidence:** 4

**Summary:**

This paper introduces VCG-Bench, a benchmark evaluating Vision-Language Models (VLMs) on generating and editing structured diagrams. It proposes a "Diagram-as-Code" approach using xml to overcome the hallucination and editability limits of pixel-based image generation. The benchmark provides 1,449 diagrams to test two tasks: Vision-to-Code (generating XML from images) and Code-to-Code (editing XML via token-efficient JSON patches). The authors evaluate 11 state-of-the-art models using executability rates, visual similarity, and LLM-as-a-judge metrics. The main findings show that open-source models currently fail at visual grounding for generation, but nearly all models succeed at precise editing once the underlying XML structure is provided.

**Compliance With Llm Reviewing Policy:**

Affirmed.

**Key Questions For Authors:**

Can you provide an empirical baseline evaluating a standard pixel-based image editing model on Task 2?

**Limitations:**

yes

**Strengths And Weaknesses:**

Strength

* Structured diagram generation and editing is a major weakness in multimodal AI. Shifting the focus from pixel-level diffusion to structured code generation is a highly relevant and pragmatic direction.
* The use of mxGraph XML is a strong choice, as it captures topology and semantics much better than SVG or HTML/CSS. Furthermore, the Code-to-Code task's use of a differential JSON patching schema is clever, token-efficient, and closely mimics real-world software engineering workflows.
* The paper evaluates an impressive array of cutting-edge models (e.g., GPT-5.2, Claude 4.5, Qwen3-VL, DeepSeek-V3.2), providing a highly current snapshot of the field's capabilities and clearly exposing the bottlenecks in open-source models.

Weakness

* A core motivation of the paper is the superiority of Diagram-as-Code over Diagram-as-Pixels for editing tasks. However, this comparison remains largely theoretical and qualitative. The paper would be significantly stronger if it empirically benchmarked a state-of-the-art diffusion editing model (e.g. nano-banana, gpt-image) the dataset to concretely demonstrate the gap in visual fidelity and instruction compliance.

---

> ### Author Rebuttal · Authors · 2026-03-30
>
> **To Reviewer TQjD**
>
> Thank you for your affirmation and detailed review. Regarding the core issue that the empirical comparison between "Diagram-as-Code" and "Diagram-as-Pixels" is lacking, we have supplemented a pixel‑based editing baseline on the Task 2 subset. The detailed response is as follows.
>
> **Weakness: Empirical Comparison between Diagram-as-Code vs. Diagram-as-Pixels**
>
> You pointed out the lack of an empirical baseline when comparing "Diagram-as-Code" and "Diagram-as-Pixels". We agree with this critique and have added a pixel editing baseline experiment on the Task 2 subset to directly address the concern that "the current comparison remains more at the theoretical and qualitative level". We will include the experimental details in the appendix.
>
> **Q1: Pixel-based Image Editing Baseline on Task 2**
>
> > *Could you provide an empirical baseline: evaluating a standard pixel-based image editing model on Task 2?*
>
> **Answer 1:**
>
> We have supplemented a pixel editing baseline experiment on the Task 2 subset to directly address your concern about whether there is an empirical gap at the dataset level between "Diagram-as-Code" and "Diagram-as-Pixels". These details will be added to the appendix later.
>
> **1. Experimental Setup.** We performed the comparison on a subset aligned with Task 2 (32 samples × 3 instructions, totaling **96** instructions; the DRFR human evaluation comprised **335** questions). The input for both approaches consisted of rendered diagram PNGs paired with the same English editing instructions. For the pixel editing side, we used **nano-banana**, and the output images were evaluated using **DRFR** human scoring per question, following the benchmark protocol. For the symbolic editing side, we used **gemini-3-pro-preview** (as in the paper) to score these sampled instances, calculating **XDRFR** automatic scores based on incremental XML/JSON patches. The following experiment directly addresses the request for an empirical comparison between "pixel editing vs. symbolic editing".
>
> **2. Quantitative Results**:
>
> | Difficulty | Instruction Count | Subquestion Count | DRFR (Pixel) |  XDRFR (Gemini)↑* |
> | ------- | --- | ---- | -------- | --------------- |
> | Overall | 96  | 335  | 0.821    | 0.940           |
> | Easy    | 32  | 50   | 0.960    | 0.983           |
> | Medium  | 32  | 97   | 0.856    | 0.969           |
> | Hard    | 32  | 188  | 0.649    | 0.867           |
>
> The results indicate that, under the setup aligned with Task 2, the pixel editing baseline is significantly weaker than the symbolic editing approach overall and across all difficulty levels, with the performance gap widening as instruction complexity increases.
>
> **3. Qualitative Observations and Editability.** Common issues observed with pixel editing models include: **text blurring or incorrect modification**, **unintended deletion or alteration of irrelevant elements**, and **difficulty in stably maintaining connections and topological relationships**. Even when instructions are superficially followed, **geometric positions, font sizes, and color schemes** often fail to meet the requirements for professional diagrams. Especially for **fine-grained local modifications** such as "**make it a little bigger**" or "**move it a little to the left**", the models frequently **ignore minor adjustment instructions** or, even if changes are made, the **magnitude does not match the target**. Furthermore, the output of pixel-based models **cannot be directly refined at the structured-object level in Draw.io-like workflows** (they can only be used as image inputs again or edited with pixel-level tools), so if the initial editing round does not meet expectations, subsequent structural refinement remains difficult. In contrast, the symbolic approach allows patches to be reapplied and rendered through the same pipeline used for evaluation, and the XML can be opened directly in Draw.io for fine-grained revision, better aligning with a realistic editable workflow.
>
> **4. Anonymous Repository.** We have organized the supplementary experimental results in an anonymous repository for the reviewer's quick reference: https://anonymous.4open.science/r/task2_pixel_baseline-F213/. Through this link, reviewers can browse **per-sample outcomes** under the **same editing instruction**, comparing **pixel vs. symbolic** outputs (rendered images and XML) to **visually verify whether edits are faithfully realized, whether fine-grained details and text remain clear, and whether edges and topology are preserved**—thus grounding the qualitative discussion and the quantitative table above. Per-file descriptions and a suggested reading order are documented in the repository **README**; we omit the per-file listing here.

---

### Decision · Program_Chairs · 2026-04-30

**Decision:**

Accept (regular)

**Comment:**

The paper introduces VCG-Bench for structured diagram generation and editing in mxGraph/Draw.io, with 1,449 diagrams across 6 domains and 15 sub-domains, two tasks (vision-to-code generation and instruction-based editing), and a task-specific evaluation suite including ESR, SCS, CodeXQA, and XDRFR.

Across the reviews, there is broad agreement that the problem is practically important, the benchmark is clearly presented, and the “Diagram-as-Code” framing is useful for editable workflows rather than screenshot-style outputs. The positive case for the paper is that it targets a real evaluation gap: structured, editable diagram generation/editing is not well covered by image-only multimodal benchmarks, and several reviewers found the benchmark design practical and likely to be useful to the community. TQjD especially liked the mxGraph choice and the differential JSON patching formulation for Task 2; GbwP and S7dz both acknowledged the practical value and broad coverage; n55j viewed the benchmark construction and experimental scope positively.

The concerns that most affected the decision is about evaluation robustness and scope. These were: the absence of an empirical pixel-editing baseline for the “Diagram-as-Code vs Diagram-as-Pixels” claim; heavy Gemini-family dependence in the data/evaluation pipeline; the fact that the main Task 1 setting uses structured captions and is therefore not pure raw-image-to-code; the limited discriminative headroom of Task 2, where many models already score very highly; and whether the paper’s novelty and scale are sufficient for a “unified benchmark” claim. The paper’s own limitations section also acknowledges restriction to mxGraph, undercoverage of non-English labels/rare styles, and the fact that executability does not guarantee semantic correctness.

TQjD: final position remains 4. The reviewer explicitly thanked the authors for the extra experiments and said they would keep the positive score.
GbwP: final position remains 4. The reviewer stated that the rebuttal addressed the main questions and provided helpful clarification, but that some minor limitations remained; they kept the original recommendation unchanged.
n55j: effective final position is 4, strengthened after rebuttal. The reviewer wrote that the concerns were fully resolved and that they would raise the score.
S7dz: final position remains 3 (Weak Reject). The reviewer stated that the rebuttal improved clarity but did not materially change their assessment, and they maintained the original score. Remaining concerns are mostly on lack of methodological contribution and concerns on dataset scale and diversity.

My suggested decision is Weak Accept. The rebuttal materially improved the paper on the main soundness questions for three of the four reviewers, and the remaining dissent is centered on dataset scope, while a small yet high quality dataset could still be useful to the community when used properly. This is still a borderline case, so the camera-ready should incorporate the rebuttal clarifications explicitly.